# Predictable recovery rates in near-surface materials after earthquake damage

Luc Illien [1] ✉, Jens M. Turowski [1], Christoph Sens-Schönfelder [1],
Clément Berenfeld[2] & Niels Hovius [1,3]

Earthquakes introduce transient mechanical damage in the subsurface, which causes postseismic hazards and can take years to recover. This observation has been linked to relaxation, a phenomenon observed in a wide range of materials after straining perturbations, but systematic controls on the recovery duration in the shallow subsurface after earthquake ground shaking have not been determined. Here, we analyse the effects of two successive large earthquakes and their aftershocks on ground properties using estimates of seismic velocity from ambient noise interferometry. We show that the relaxation time scale is a constant that is an intrinsic property of the substrate, independent of the intensity of ground shaking. Our study highlights the predictability of earthquake damage dynamics in the shallow subsurface and also in other materials. This finding may be reconciled with existing state variable frameworks by considering the superposition of different populations of damaged contacts.

Earthquakes can cause catastrophic effects at the Earth's surface, which range from urban damage to physical changes in the landscape and hydrosphere[1,2]. For instance, landsliding[2], aquifer permeability[3] and river discharge[4] can peak after seismic strong ground motion, before eventually returning to pre-earthquake levels. Although they are transient, these effects can persist multiple years, increasing risk after major earthquakes[1].

Transient geomorphic and hydrologic phenomena are observed at the surface, but they suggest a change in the mechanical properties of the Earth's shallow subsurface in the aftermath of earthquakes. It has been found that earthquakes induce a drop in subsurface seismic velocity, which is followed by a recovery on time scales ranging from a few days to several years[5–12]. The recovery dynamics resemble universal relaxation[13], a phenomenon observed in rocks[14–16], granular media[17], concrete[18] and civil structures[19]. In the field, seismic velocity changes can be obtained through correlations of ambient seismic noise, a technique called seismic interferometry. The relaxation timescale observed with this method has been shown to correlate with the duration of enhanced postseismic landsliding in epicentral areas[20]. This suggests that the latter—and possibly other transient surface processes - originates from transient damage represented by the relaxation dynamics of the shallow subsurface. Hence, a physically

based model with clear observables, to predict and constrain the duration of the recovery could strategically inform the hazard mitigation after earthquakes, and guide the monitoring of affected buildings, landscapes and fault zones.

The relaxation of earthquake-induced damage in subsurface materials is commonly analysed by fitting an exponential decay function to observed seismic velocity changes, with a single timescale[21,22]. This empirical method has yielded numerous relaxation timescales ranging from a few days to several years[10,20] but does not give insight into the main control of this duration. Moreover, the recovery of the seismic velocity changes exhibits a logarithmic evolution, which the exponential fit fails to capture[22]. It is clear that current practice provides an insufficient basis for reliable prediction of post-seismic recovery.

Motivated by this lack of physical framework and predictive power, models based on contact mechanics and friction physics have been recently used to explain observed velocity changes[23,24]. In these models, the rate of change of the velocity $dv/dt$ is a direct function of the current state velocity $v(t)$. If these expressions are appropriate, then they permit the prediction of the relaxation duration from seismic velocity data. To date, seismic monitoring studies have mostly considered the impact of singular earthquakes, ignoring the potential importance of concatenated relaxations due to multiple seismic

[1]GFZ Helmholtz Centre for Geosciences, Potsdam, Germany. [2]Universität Potsdam, Department of Mathematics, Potsdam, Germany. [3]Universität Potsdam, Department of Geosciences, Potsdam, Germany. ✉e-mail: lillien@gfz-potsdam.de

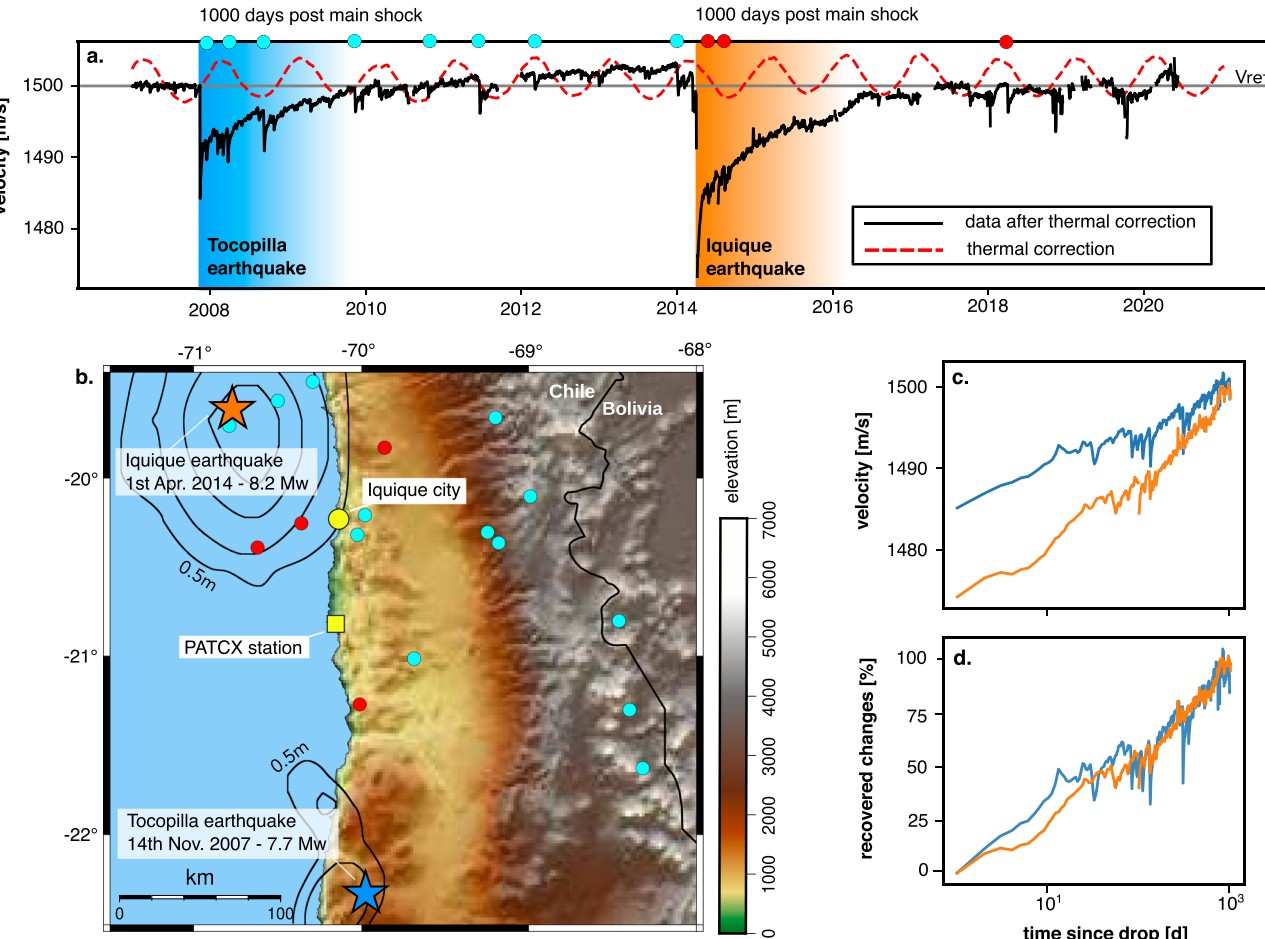

**Fig. 1 | 13 years of seismic velocity data for Station PATCX in Chile, including transient changes due to the Tocopilla and Iquique earthquakes. a** The black-line indicates the relative seismic velocity changes plotted against a reference velocity of 1500 m/s (horizontal grey line). The red line shows the correction for thermal strain due to recorded temperature fluctuations. Dots at the top mark the occurrence of major aftershocks. **b** Map showing the study area (topography from ETOPO 2022[48]) with stars indicating the epicenters of the main shocks. The black isolines show the slip distribution of the main shocks[49,50] (0.5–1 m for the Tocopilla event and 0.5–3 m for the Iquique event). Red and cyan dots indicate the aftershocks inducing velocity drops (full list in Table S1). **c** Plot of seismic velocity recovery during the first 1000 days following the main shocks on a logarithmic time-axis. The blue and orange lines are for the Tocopilla and the Iquique earthquake episode respectively. **d** Plot of seismic velocity recoveries normalised by the amplitude of the respective initial velocity drop.

events. This hampers robust testing of models based on friction physics, and precludes their use in prediction and explanation of postseismic relaxation.

Here, we aim to estimate daily seismic velocity changes in response to multiple earthquakes to improve current models of postseismic subsurface recovery in natural geological systems. The seismic station PATCX in the Atacama desert of Chile (Fig. 1ab) was chosen for this study for three reasons. First, the site has experienced two major earthquakes, the 2007 Mw 7.7 Tocopilla and the 2014 Mw 8.2 Iquique, each with numerous aftershocks, giving the opportunity to study multiple episodes of cumulative damage and relaxation of the subsurface on relevant timescales. Second, previous work has demonstrated that the station records allow highly stable measurements of very small velocity changes[22,25–27]. And third, the hyper-arid setting precludes significant effects of changes in water content (-1–5 mm/year of precipitation[28]). Using a 13-year time series of seismic velocity changes, we go beyond the former studies conducted at PATCX to show that the subsurface response to multiple individual seismic events can be predicted using a single value of recovery timescale (in this study  -1.5–2 years) independent of the various ground shaking intensities.

## Results and discussion
### 13 years of seismic velocity changes induced by earthquake damage

The relative seismic velocity changes at the PATCX site are controlled by all forcings that modify rock properties in the subsurface. Notably, thermal strain at depth due to the seasonal temperature variations in the Atacama desert possibly masks the influence of earthquake damage[25]. We modelled these thermal effects with a simple 1D heat diffusion approach (Methods) to obtain residual relative velocity changes. In the absence of significant hydrological influences, we assume that these residual changes represent the response to seismic ground shaking (Fig. 1a). Then, we scaled the relative seismic velocity measurement to the reference value for the local shear velocity $V_S$ =1500 m/s[27]. The resulting time-series shows velocity drops of 15 and 35 m/s for the Tocopilla and Iquique earthquake, respectively. Each was followed by a recovery period of three to four years, the longer after the Iquique earthquake. Additional drops and associated recoveries due to aftershocks and smaller earthquakes are evident, particularly in the aftermath of the Tocopilla earthquake. Drop amplitudes have been shown to be proportional to the peak ground velocity at PATCX and other sites[22,25].

We emphasize the log-linear behaviour of the post-seismic velocity recovery by plotting the first 1000 days of the recovery phases following the two large earthquakes on a logarithmic scale (Fig. 1c). For three orders of time magnitude, the recoveries after both large earthquakes appear linear in this space, with some deviations due to aftershocks. The consistency of the slope despite these additional perturbations suggests that the recovery timescale may be predictable by measuring the rate of velocity change in the first weeks after an earthquake, provided that other effects (e.g., hydrology) can be excluded.

We normalised the recoveries by considering the percentage of recovered change relative to the amplitude of the velocity drop (Fig. 1d). This normalisation collapses the time series for the two earthquakes onto a single line. This indicates that, although despite causing different drop magnitudes, both earthquakes induced a relaxation behaviour that had a similar timescale of recovery over the first 1000 post-seismic days. The co-seismic velocity reductions were approximately halved in the first 200 days of recovery following the main shocks. Aftershocks with smaller attendant velocity drops did not have a significant impact on the total relaxation duration. We estimate that the full recovery of the seismic velocity after Tocopilla and Iquique earthquakes would take about 2.5–3 years (Fig. 1a, d).

## A constant recovery timescale after seismic events

We observed that the velocity immediately prior to ground shaking, as well as the recovery timescales were very similar after the Tocopilla and Iquique earthquakes. This suggests that the subsurface relaxation is a function of the medium properties rather than the initial intensity of that ground shaking, and that the subsurface has an intrinsic recovery timescale. This hypothesis can be explored further by analysing the recoveries following aftershocks, extending the range of pre-drop velocity conditions prior to earthquakes. The first 10 days of recovery after each seismic event can be visualised and normalised by the pre-drop velocity level, here represented by the velocity on the day prior to the aftershock (Fig. 2a). A 10-day window was chosen to minimise the potential superposition of recoveries associated with previous aftershocks. In this time, recovery to the pre-aftershock baseline ranged from 20–30% for some aftershocks to 100% for others.

A correlation appears to exist between the seismic velocities just before aftershocks and the subsequent recovery duration, with faster recovery for drops with lower pre-drop velocity (Fig. 2a). We quantified this trend by fitting the 10-day recovery histories with a universal relaxation function[29] (Methods, Fig. S4b). This yields a likely maximum relaxation timescales $\tau_{max}$ for each aftershock, where the value of $\tau_{max}$ represents the recovery time to the seismic velocity observed before the drop. The distribution of the inferred timescales $\tau_{max}$ for the aftershocks in our time series ranges from ~8 to ~3000 days and can be described by an exponential relationship ($R^2 = 0.86$) with the pre-drop velocity. This scaling suggests that at low velocities relative to a baseline in which the velocities seem to be at steady state (here ~1500 m/s), the subsurface at PATCX may be more resilient with systematically shorter recovery times.

Using this fitted exponential relationship, we built a synthetic seismic velocity time-series. We measured all the velocity drops in the data, calculated their recovery timescale with their pre-drop velocity, and superposed them linearly, creating one time-series with multiple relaxations. Even if this synthetic time-series does not directly fit the full measured dataset, there is a good overall agreement with the instrumental data (Fig. 2c, Nash-Sutcliff efficiency NSE = 0.68, Methods). However, we can not rule out that the scaling of the pre-event velocity and the maximum relaxation timescale $\tau_{max}$ shown in Fig. 2b may be due to the co-evolution of multiple recoveries characterized by a single constant recovery time scale after all earthquakes[30]. In this scenario, the superposition of the same timescale of recovery

following subsequent aftershocks could lead to an apparent scaling between the pre-drop velocity and the recovery timescale.

We test the hypothesis of a constant recovery time scale after all seismic events using values for $\tau_{max}$ ranging from one to four years (Fig. 2d). The ability of this method to predict the $dv/v$ data outperforms the former approach with NSE values above 0.76 for $\tau_{max} \sim 1.5$-2 years (Fig. 2e). For all models with constant $\tau_{max}$, we also plot the recoveries normalised by the initial drop amplitudes to show the emergence of the apparent scaling between the pre-drop velocity and the recovery timescale (Fig. S5). This suggests that in the range of confining pressure and dynamic strain of our study, a single maximum recovery timescale $\tau_{max}$ value can be used to model the entire time-series. At PATCX, the total recovery duration observed after the Iquique and Tocopilla earthquakes is about three years but is in reality the result of multiple seismic events (the main shocks and their aftershocks) exciting a fundamental recovery timescale of 1.5-2 years. We note that a direct fit of the full average recovery after the main shocks (including their aftershocks sequences) would lead to a spurious duration $\tau_{max}$ of about 10 years (Fig. S4a).

The remaining offsets between the data and the modeled time-series (particularly visible in the 2012–2014 period) may be due to the limitation of the temperature correction, a new seismic velocity base level following the Tocopilla earthquake or a draw-down of groundwater levels in the Atacama region[31], which could affect the velocity changes[32,33].

## Which physical framework can describe near-surface earthquake damage ?

Our observations suggest that the timescale of seismic velocity recovery at a site is constant and can be predicted from the analysis of former relaxations at a site and aftershock sequences. This feature is not directly tied to existing physical equations for relaxation. Most existing physical and conceptual models associate the relaxation dynamics in seismic velocity in proportion to a 'state variable' representing the dynamics of the dominant type of contact or void embedded in a medium undergoing deformation. In these frameworks, the evolution of the variable of interest, here $dv/dt$, corresponds to a single value of the variable itself $v(t)$ through a recovery function. This is the case in the rate and state friction framework[34] and models that involve contact dynamics[35] and creeping mechanisms to fit relaxation data[23,24,36]. Field studies often fit such models based on one relaxation event. Our dataset, comprising 18 consecutive drop-and-recovery cycles does not support this state variable approach: the phase portrait plot between the rate of velocity change $dv/dt$ and the velocity $v(t)$ shows very different phase locations after each seismic event (Fig. S6) i.e., the same rates of variation $dv/dt$ can be observed with different state velocity $v(t)$. This indicates that the parameters of the fitted models would need to be changed at each drop and recovery cycle. Still, we suggest how classic state variable functions can be reconciled with our observations in the following.

In the field, seismic velocity measurements probe a subsurface volume that contains many types of contacts and flaws[16,29,37]. To account for their different dynamics, we assumed that all of these structures obey their own state variable equations, each with their timescale of relaxation. For instance, crack populations can be defined by their aspect ratios, which determine their time of closure under a specific load[38,39]. We used a generic state variable in a phenomenological model (Methods) that can characterize different processes, each with their own timescale (Fig. 3a). The simple mean of all these state variable dynamics yields a synthetic time-series with the same features as our seismic velocity observations: A logarithmic recovery of velocity after each drop at a rate dependent on the pre-drop velocity (Fig. 3b, c). Therefore, the superposition of different timescales in a sequence of perturbations can reproduce the dynamics we observed at station PATCX, which is dominated by the longest relaxation timescale

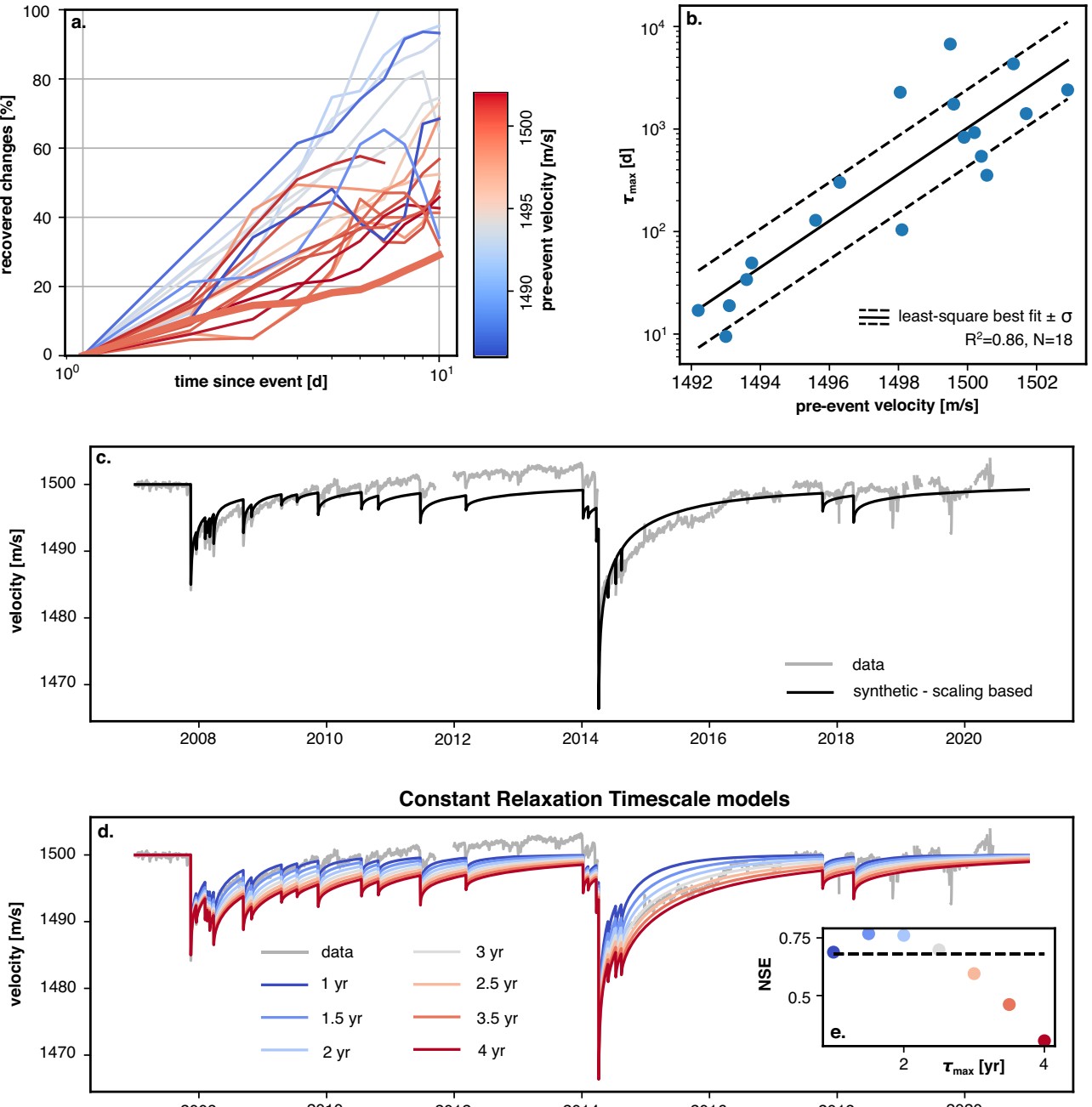

**Fig. 2 | Pre-drop velocity level as a control on the relaxation timescale and corresponding synthetic time-series. a** Recoveries after individual seismic events, normalised by the amplitude of the drop. The colour indicates the relevant pre-drop velocity. The thick line shows the early average recovery observed after the Tocopilla and Iquique main shocks. **b** Fitted maximum recovery timescales for the recoveries shown in (**a**) (N = 18). The black lines show the best fit exponential scaling ($\tau_{max} = \exp(a \cdot V_0 + b)$ for a = $5.2226.10^{-1}$ and b = $-7.7645.10^2$) and its standard deviation. **c** Synthetic velocity time-series built by superposition of relaxations obtained with the relationship shown in **b** The grey line shows the instrumental data. **d** Synthetic velocity time-series built by superposition of constant timescales of relaxation, indicated by the colour of the curves. **e** Nash-Sutcliff efficiency (NSE) for all tested models with constant $\tau_{max}$, the colours correspond to relaxation timescales shown in (**d**). The dashed line is the NSE value for the $dv/v$ model obtained with the scaling on (**c**).

in the system $\tau_{max}$ (Fig. 3). This combined contribution of different states of evolution is key to reconciling observations and physics. At the reference velocity value (in our study around 1500 m/s), the density of contacts in the rocks is constant and the system is at steady state. When the contacts are disturbed, they all enter a metastable state and recover at different rates. As the general reference velocity is approached, only the contact populations with the longest recovery times remain activated, while populations with a faster recovery have stabilized (Fig. 3). Assuming that the internal structures in the

subsurface at a site are not fundamentally changed (no new types of contacts or flaws), we expect that the recovery of seismic velocity due to earthquakes can be predicted based on this theory and is the result of multiple constant timescales of relaxation. For higher state of damage (e.g after slip on fault interfaces), the timescales of recovery may change[40–42]. The boundary between these regimes is yet to be defined in future studies.

Building upon the observation of universal relaxation in the lab[13], our study shows that after an earthquake, the mechanical properties of

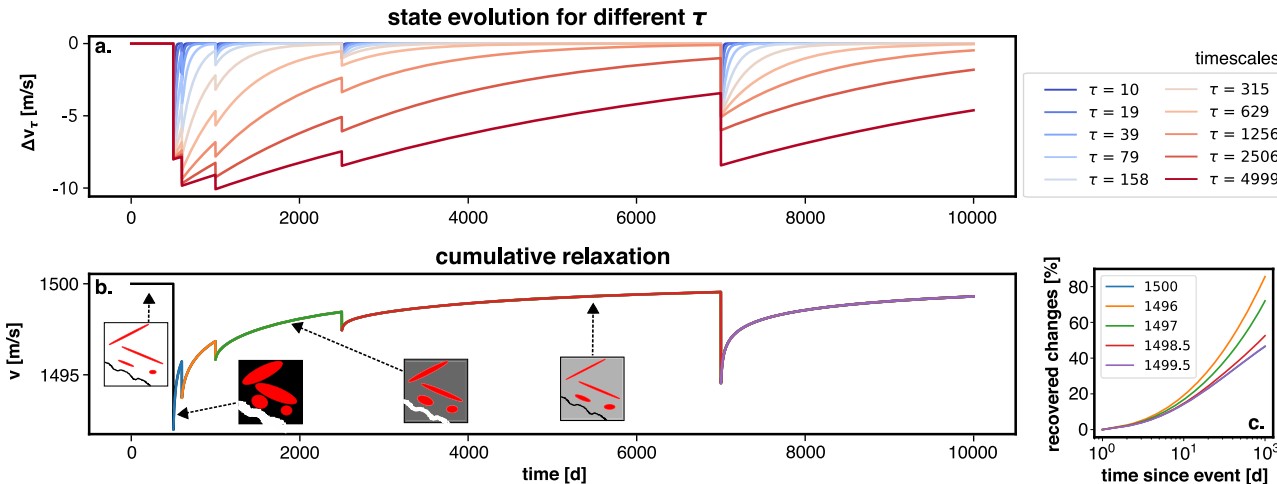

**Fig. 3 | Superposition of different state variable dynamics and average relaxation. a** Each curve shows the evolution of a hypothetical state variable with a distinct characteristic timescale. The timescales are shown in the legend. **b** Average evolution of all state variables shown in (**a**). The colours on the line stand for the recoveries after each drop (also shown in (**c**)). The schematics indicate different types of cracks/voids with different closure times corresponding to different state variables in (**a**). The background colour indicates the relaxation of the stress state in the bulk. **c** The first 100 days of the relaxation phases shown in (**b**). are plotted on a log-linear scale and normalised by the amplitude of the initial drop. The colours indicate the value of the velocity before the drop onset on both plots (**b** and **c**).

a subsurface medium control its relaxation behaviour and timescale of recovery, while the ground shaking intensity only controls the amplitude of the velocity change or the magnitude of damage. If this recovery behaviour is valid for near-surface geomaterials, then it may also be tested for man-made structures[19]. Understanding the link between specific types of flaws and relaxation timescales can lead to a targeted material test for management of the response to damage. For earthquake source physics and in particular for dynamic triggering of fault interfaces[43], one could test if our findings apply e.g. the propensity to initiate aftershocks would be controlled by the interface properties during fault healing[44] rather by the dynamic strain amplitude. A representative relaxation timescale may also allow an estimation of the propensity to failure of steep topographic slopes in an earthquake epicentral area[20] and therefore an assessment of the feasibility and timeliness of reconstruction after large earthquakes. This can reduce lingering post-seismic risk, and minimize spurious investment. Our results should also be tested in presence of pronounced anisotropy in which different recovery duration could be observed with variable seismic polarizations. Finally, we propose that measuring the rate of velocity change over a period immediately after an episode of strong ground motion and before any damaging aftershocks, can yield early constraints on the likely pace of recovery of substrate damage. This early estimation can be based on measurements from small seismic events to constrain the likely maximum relaxation $\tau_{max}$ at a specific site and then upscaled for larger earthquakes and their aftershock sequences.

## Methods
### Seismic interferometry
We computed the 6 self-correlations (EE, NN; ZZ, EN, EZ, NZ) of the PATCX station after filtering the traces in the 3–6 Hz band and muting seismic segments in which the amplitudes do not follow a Gaussian probability distribution[27]. We calculated one-hour long cross correlation functions that we stacked to obtain daily cross-correlations. To measure the velocity changes, the stretching technique[45] was applied in the 7–16 seconds window of the daily correlation functions. In a last step, the results for the 6 self-correlations are averaged.

### Temperature-correction
We first quantified the phase of the periodic temperature (taken at the Iquique airport, 30 km from the station) and $dv/v$ seasonal variations.

Temperature data were first smoothed with a Savitzky-Golay filter of 30 days length and order 3. The Savitzky-Golay filter smooths the data using a convolution with polynomials instead of a boxcar function (a simple moving average is a Savitzy-Golay filter of order 1). This has the advantage of retaining more high-frequency information in the time-series preserving more potential data peaks in the smoothing window[46]. To approximate a periodical variation, we calculated the average temperature and velocity cycle by stacking the yearly temperature and $dv/v$ time-series and fitting the average stack with a sinusoidal function (Figure S1). The fitting of the velocity data yields a first empirical correction of the thermal-induced velocity changes (red line in Fig. S3). The phase lag between the temperature and the velocity is equal to -19 days.

We assumed that the velocity changes originate from the thermal strain observed at one depth $z$ due to thermal diffusion in a subsurface layer of thermal diffusivity $\kappa$. For a periodic temperature oscillation at the surface (of period $\omega$), the phase lag for the temperature $T$ at depth $z$ can be retrieved with the following equation[47]:

$$T = T_0 + \Delta T \exp\left(-z\sqrt{\frac{\omega}{2\kappa}}\right) \cos\left(\omega t - z\sqrt{\frac{\omega}{2\kappa}}\right). \qquad (1)$$

In this relation, one can see that the phase lag is equal to $-z\sqrt{\frac{\omega}{2\kappa}}$. From the phase difference calculated before and after setting the period $\omega$ to one year, we obtain the value of the ratio $z^2/2\kappa$, which determines the system behaviour. Assuming now a generic thermal diffusivity of $\kappa = 1.10^{-2}$ cm²/s, we can model the velocity changes from the temperature observed at the characteristic depth of the ratio ($z = 1.25$ m). Plugging these values in a 1D thermal diffusion numerical simulation (computed with a forward Euler scheme) and using the surface temperature time-series (Fig. S2a), we obtained the temperature depth profiles shown in Figure S2b. We extracted the temperature at depth $z = 1.25$ m and scale it to the velocity changes through a factor $\alpha$, which represent the expansion coefficient of the rocks in the subsurface ($\alpha$ -0.053). This scaling is done on the velocity data of 2007 prior to the Tocopilla earthquake. The velocity time-series from this simulation is shown in Figure S3a.

The retained temperature model for the paper is taken as the average between the two modeled $dv/v$ time-series (Fig. S3a). Indeed, we combine the best of both corrections as **1.** the numerical correction is based on the temperature time-series and may have the best

correction for the phase of the temperature-induced velocity changes and **2.** the amplitudes of the velocity changes is more consistent with the sinusoidal correction.

## Measurement of $\tau_{max}$

We used the expression of Snieder et al., to find the maximum relaxation timescale $\tau_{max}$, which controls the duration of the recovery phase after earthquakes:

$$R(t) = \int_{\tau_{min}}^{\tau_{max}} \frac{1}{\tau} e^{-(t-t_0)/\tau} \, d\tau. \tag{2}$$

From equation (2), one can see that function $R(t)$ is therefore a superposition of exponential recoveries characterised by timescales $\tau$. From a previous publication at the PATCX field site[27], we know that the minimum relaxation timescale $\tau_{min}$ is inferior to one minute. At the daily time resolution of the retrieved velocity time-series of this study, we also do not resolve $\tau_{min}$. Therefore, we fixed $\tau_{min}$ to one day and scaled $R(t)$ to the velocity measurements and find the best fitting $\tau_{max}$.

We first quantified the relaxation time of the Iquique and Tocopilla earthquakes, assuming from Fig. 1d that both earthquakes are followed by the same recovery timescale. Therefore, we averaged the first 1000 days of relaxation of both time-series (Fig. S4a) and interpolated the resulting recovery on a logarithmic x-axis. The latter step gives an equal weight to the early relaxation times in the fitting process. Finally, we fit and scaled the expression $R(t)$ and obtain a best fitting value of $\tau_{max} = 3887$ d.

For each aftershock-induced recovery, $\tau_{max}$ was obtained by fitting the first 10 days of relaxation (Fig. S4b) without the interpolation step. We also report the $\tau_{max}$ values corresponding to $\pm 10\%$ of the variance of the best fitting value (orange and green lines in Fig. 2b).

## Estimating the performance of the synthetic seismic velocity time-series

For comparing the efficiency of the two models, we used the Nash-Sutcliffe efficiency (NSE) coefficient. The coefficient estimates the performance of a model at predicting a time-series relative to the mean of the time-series. This is calculated as

$$\text{NSE} = 1 - \frac{\sum(v - v_m)^2}{\sum(v - \bar{v})^2}. \tag{3}$$

In this equation, $v$ is the time-series that we want to predict, $v_m$ is the predicted time-series from a model and $\bar{v}$ is the mean of the observed time-series. We obtained NSE $= 0.68$ for the model featuring the exponential scaling and NSE $= -2.61$ for the model featuring the constant timescale of relaxation. The negative coefficient indicates that the model with the constant timescale of relaxation is less efficient at predicting the data than the mean of the seismic velocity changes.

## State variable theory

We look for an equation that is homogeneous to the velocity changes and choose a form that is mathematically viable with the logarithmic evolution of the recovery. We assume full recovery to the pre-earthquake seismic velocity state (No generation of new contacts/flaws). A general equation for the velocity variations $dv/dt$ can be expressed as

$$\frac{dv}{dt}(t) = A\exp[-Bv(t)] - C, \tag{4}$$

where $A$, $B$ and $C$ are constants with $A$ and $C$ having the same dimension than $dv/dt$, and $B$ having the dimension of $1/v$.

Note that the expression (4) has the same form than Arrhenius equation[29], used to model thermal-induced processes and creep rates.

The constant $A$ is generally interpreted as the number of molecular collisions characteristic to the physical system at play while $B$ is linked to the activation energy of the reaction. We have added $C$ to impose a limit on the recovery at which the rate of velocity changes does not change anymore ($dv/dt = 0$) so that the recovery is complete. Based on the dimension of the constants, we further parametrise (4) to highlight a timescale $\tau$: letting $B = 1/v^*$, $C = v^*/\tau$ and $A = v^* e^{v_\infty/v^*}/\tau$, we get

$$\frac{dv_\tau}{dt}(t) = \frac{v^*}{\tau}\left(\exp\left[-\frac{v_\tau(t) - v_\infty}{v^*}\right] - 1\right), \tag{5}$$

in which $v_\infty$ is the steady state velocity value in the absence of earthquakes, $v^*$ is a velocity modulating the prefactor and the exponential term and $v_\tau$ is the velocity associated with a certain relaxation time $\tau$. We consider several contacts or flaws with different characteristic timescales $\tau$, which affect the velocity changes. The total observed velocity change $v(t)$ is a function of these voids. The contribution of each $v_\tau$ to the total observed velocity changes $v(t)$ depends on the density of the associated voids/contacts and their respective sizes. For the sake of simplicity, we take in this paper the average of all $v_\tau$ dynamics present in the system:

$$v(t) = \frac{1}{N}\sum_{i=1}^{N} v_{\tau_i}(t) \tag{6}$$

To show the logarithmic evolution of our framework, we now derive the expression of the solution of (5) for initial condition $v_0 = v_\tau(0) < v_\infty$, and denote by $\Delta = v_\infty - v_0$ the size of the initial velocity drop. Letting $w(t) = \exp(-(v_\tau(t) - v_\infty)/v^*)$, there holds

$$\frac{dw}{dt}(t) = -\frac{1}{v^*}\frac{dv_\tau}{dt}(t) \times w(t) = -\frac{1}{\tau}(w(t) - 1) \times w(t) \tag{7}$$

which in turn yields

$$\int_0^t \frac{dw}{w(w-1)} = -\int_0^t \frac{dt}{\tau} = -\frac{t}{\tau}. \tag{8}$$

The LHS of the above display can be expressed as

$$\int_0^t \frac{dw}{w(w-1)} = \int_0^t\left(\frac{1}{w-1} - \frac{1}{w}\right)dw = \left[\log\left(\frac{w-1}{w}\right)\right]_0^t = \log\left(\frac{w(0)(w(t)-1)}{(w(0)-1)w(t)}\right) \tag{9}$$

Letting $\xi_0 = (w(0) - 1)/w(0)$, we finally find that $w(t) - 1 = \xi_0 e^{-t/\tau}w(t)$, which immediately gives $w(t) = (1 - \xi_0 e^{-t/\tau})^{-1}$. Expressing $w$ back as a function of $v_\tau$ finally yields

$$v_\tau(t) = v_\infty - v^*\log\left(1 - (1 - e^{-\Delta/v^*})e^{-t/\tau}\right). \tag{10}$$

Using this framework, we computed the synthetic relative seismic velocity variations shown in Fig. 3, which span a period of 10 000 days (-27 years). First, we arbitrarily define a logarithmic range of relaxation times $\tau$ from 10 to 4999 days (all values in Fig. 3). For each of these $\tau$ values, the goal is to estimate a time-series $v_\tau(t)$ that would correspond to a relaxation phenomenon defined by its timescale $\tau$. We fix $v_\infty$ to 0 to obtain the amplitude of the velocity perturbations ($v_\tau(t)$ to $\Delta v_\tau(t)$) and defined 5 consecutive velocity drops $\Delta$ (−8, −2,−1,−1 and −5 m/s at respectively 500, 620, 1300, 5000, 7000 days in the time-series). Finally, we fix the value of $v^*$ to 1000 m/s for all $\Delta v_\tau(t)$. Using these values, equation (5) can be numerically integrated to obtain the time-series shown in Fig. 3a.

In a last step, we averaged all $\Delta v_\tau(t)$ perturbations following equation (6) and added the value $v_\infty = 1500$ m/s (reference value at our field site) to the time-series to obtain $v(t)$ as plotted on Fig. 3b.

## Data availability

Data for the station PATCX are in open access and can be downloaded at the GEOFON repository (https://doi.org/10.14470/PK615318). The compiled results can also be found at the following Figshare repository: https://doi.org/10.6084/m9.figshare.28071695).

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

## Acknowledgements

L.I. would like to acknowledge Nicolas Brantut for insightful discussions.

## Author contributions

The conceptualization of this work and the research were performed by L.I. under the supervision of J.T., C.S., and N.H. C.B. assisted L.I. for the theoretical section. The original draft was written by L.I. with further editing and reviewing by all the other co-authors.

## Funding

## Competing interests

The authors declare no conflicts of interest.
