## [Transparent Peer Review file · Nature Communications]

Predictable recovery rates in near-surface materials after earthquake damage

Corresponding Author: Dr Luc Illien

Version 0:

Reviewer comments:

Reviewer #1

(Remarks to the Author)

The comments are attached as Pdf file

Reviewer #2

(Remarks to the Author)

see pdf

Reviewer #3

(Remarks to the Author)

This study investigates the recovery process in near-surface material after earthquakes using the ambient noise correlations. The authors computed the ambient noise correlations at a single station in Chile and estimated seismic velocity changes over a 13-year period. Co-seismic velocity drops related to the 2008 Tocopilla earthquake, the 2014 Iquique earthquake, and their aftershocks were observed. By analyzing the co-seismic velocity changes for each earthquake as recovery changes, the authors showed that the relaxation time scale is controlled by the mechanical properties of a subsurface medium. The authors interpreted the recovery process by modelling the velocity changes using the state variable theory. The topic discussed in this study is very interesting, and this work could be of significance to the field. However, the manuscript was a little difficult to follow. Some clarifications will help to make it more understandable to readers. The submitted manuscript did not include line numbers, so I am commenting for each topic. Sorry if this is confusing.

Main comments:

Seismic interferometry:

What is the depth sensitivity of waves within the frequency band used in this study?

In the high-frequency band, body waves are expected to contribute to the noise correlations. In this case, the depth may exceed the shallow structure you are targeting. Is it possible to discuss changes in the near-surface materials even if body waves are assumed to dominate in the noise correlations?

Temperature-correction:

Please clarify which observation period was used for scaling temperature-induced dv/v . What was the scaling value estimated to be? And is that value reasonable compared to previous studies?

The temperature model for paper is described as the average between two modeled dv/v . Is this averaging of different models appropriate? If the scaling factor (α) changes with time, how does this affect the temperature correction? I think it is possible that the sensitivity of dv/v to the temperature changes due to the large earthquakes.

Figure 1a and state variable theory:

In this study, a state variable model is used with $dv/dt=0$ as the recovery state. However, looking at Figure 1a, the seismic velocity change seems to be larger than the pre-drop value (e.g., the velocity from 2012 to 2014 after the Tocopilla earthquake). How should this be interpreted?

Figure 2c:

An offset is visible between the observed velocity change and the model. What evidence supports attributing this offset to temperature correction?

Please clarify the concept of a "new persistent velocity level." How does this relate to the statement that dv/v after the Tocopilla earthquake recovered to the pre-drop level?

Page 4 "all contacts are at steady state and the system does not evolve.":

What specific situation is referred to when stating the system does not evolve? Does this imply that the number and orientation of cracks remain constant?

How might the generation of new contacts or flaws by ground shaking, or changes in seismic structure anisotropy due to large earthquakes, affect velocity changes? These factors appear to be excluded from the current model.

Figure S5:

Please explain the term "phase portrait" in this context. While the figure shows different dv/dt values for the same $v(t)$, it is unclear how this reconciles the state variable argument with observations.

Figure 3a:

The determining v^* , v_0 , and v_∞ when plotting this figure is unclear. A more detailed explanation how you created the figures would enhance reader understanding.

Additional comments:

Caption of Figure 2:

Please consider using a center dot instead of asterisk to denote multiplication. Also, I think you should consider how many significant digits are needed.

Temperature-correction:

As many readers may be unfamiliar with the Savitzky-Golay filter, please briefly explain its purpose and advantages over a simple moving average.

Line 4 of the section ("Which physical~"):

"of of interest" -> "of interest"

Version 1:

Reviewer comments:

Reviewer #1

(Remarks to the Author)

The comments are attached in the PDF file.

Reviewer #2

(Remarks to the Author)

The authors have done good work to revise the manuscript. They have addressed the questions of all reviewers and the paper's message is now clearer. This is a nice piece of work. I have only one outstanding issue and a few minor points.

The main issue relates to this statement, found in the abstract and elsewhere. "We show that the relaxation time scale is a function of the state of the substrate at the time of seismic perturbation, rather than the intensity of ground shaking." In fact, while I see their point but while it holds for the data shown in fig. 1 it cannot be true in an absolute sense for all states of shaking induced damage. Certainly, if the state of damage is sufficient recovery will take longer. This fact is also clear when one considers data on frictional strength recovery –frictional healing. In studies of healing one observes that the degree of strength recovery scales with time, and this applies not only to frictional strength but also to elastic properties including seismic wave speed (for example Ref 1 below). Here, one sees that the longer we wait the more the wave speed increases. I do not think that this contradicts their results, but rather adds to it and perhaps suggests a range of behaviors. Having read their rebuttal, I believe that our differences are not fundamental but rather a matter of degree. So, I would be satisfied if they used some qualification to these statements.

Other and minor points.

A. Figure 1: Dots at the top mark the occurrence of major aftershocks.

B. Yes, I see that you have added references to geophysical observations. But for me, there are still too many statements that are unqualified or insufficiently tied to previous observations. For example, in the Introduction, this sentence should also include reference to a few of the studies below. "It has been found that earthquakes induce a drop in subsurface seismic velocity, which is followed by a recovery on time scales ranging from a few days to several years." The references they have now (5-9) are valid but do not include basic studies of seismic damage and post-seismic recovery. I do not discount their references to 11-13, but there are important field studies that connect strongly to their work and should be included.

C. Why is it called 'pre-drop' velocity? Wouldn't initial velocity be better?

1. Kaproth, B. M., and C. Marone, Evolution of elastic wave speed during shear-induced damage and healing within laboratory fault zones, *J. Geophys. Res. Solid Earth*, 119, 10.1002/2014JB011051, 2014.
2. Rivière, J., Candela, T., Suderi, M.M., Marone C., Guyer, R., and P. Johnson, Dynamic acousto-elasticity in Berea sandstone: influence of the strain rate. *J. Acoustical Soc. Am.*, 134, 10.1121/1.4831394, 2013.
3. Cochran, E. S., Li, Y.G., Shearer, P. M., Barbot, S., Fialko, Y. and J.E. Vidale, Seismic and geodetic evidence for extensive, long-lived fault damage zones. *Geology*, 2009.
4. Li, Y.G. Vidale. J. E., Aki, K. Xu, F. and T. Burdett, Evidence of shallow fault zone strengthening after the 1992 m7.5 Landers, California, earthquake, *Science*, 1998.
5. Vidale, J. R. and Y.G. Li, Damage to the shallow Landers fault from the nearby Hector Mine earthquake, *Nature*, 421, 2003.
6. Tinti, E., Scuderi, M. M., Scognamiglio, L. Di Stefano, G., Marone, C., and C. Collettini, On the evolution of elastic properties during laboratory stick-slip experiments spanning the transition from slow slip to dynamic rupture, *J. Geophys. Res. Solid Earth*, 10.1002/2016JB013545, 2016.
7. Rivière, J., Pimienta, L., Scuderi, M., Candela, T., Shokouhi, P., Fortin, J., Schubnel, A., Marone C., and P. A. Johnson, Frequency, pressure and strain dependence of nonlinear elasticity in Berea sandstone, *Geophys. Res. Lett.*, 2016.

Reviewer #3

(Remarks to the Author)

Review of "Predictable recovery rates in near-surface material after earthquake damage", submitted by Luc Illien, Jens M. Turowski, Christoph Sens-Schönfelder, Clement Berenfeld, and Niels Hovius

The authors politely responded to my previous reviews. I would like to ask an additional question regarding the model of velocity time series based on state variable theory. According to the authors' response, it seems that the effect of velocity anisotropy due to large earthquake does not need to be considered in this field. However, could individual large earthquakes generate new cracks and flaws? If so, might this introduce discrepancies between the model and the data presented in Figure 2?

Furthermore, this study seems to focus solely on the effects of dynamic stress. Could static stress also influence structural changes? The spatial distribution of static stress, whether expansive or contractive, likely varies for each earthquake at the observed location. Would this variability affect the recovery process of cracks or flaws?

First line of page 2:

seismic station station -> seismic station

Version 2:

Reviewer comments:

Reviewer #1

(Remarks to the Author)

I am very grateful to the authors for providing a detailed response to my comments. This paper demonstrates that stacking recovery processes with different recovery times can better predict the recovery process of near surface wave velocity. The error between observation and simulation (i.e. the Nash Sutcliffe efficiency value mentioned in the article) decreased from 0.76 to 0.68. This result is very interesting and can deepen our understanding of the slow dynamics of geotechnical materials. Therefore, I agree to the publication of this paper.

Reviewer #2

(Remarks to the Author)

The authors have now responded fully to my reviews and, in my opinion, to those of the other reviewers. This is a very nice paper and it is ready to be published.

Reviewer #3

(Remarks to the Author)

The authors have addressed all my comments and I am happy to recommend publication of this article.

Author’s Response to Reviews of

Predictable recovery rates in near-surface materials after earthquake damage

Luc Illien, Jens Turowski, Christoph Sens-Schönfelder, Clement Berenfeld, Niels Hovius
Nature Communication,

RC: Reviewer’s Comment, AR: Author’s Response, □ Manuscript Text

1. Reviewer 1

RC: *I have carefully reviewed the manuscript titled “Predictable recovery rates in near-surface materials after earthquake damage” by Luc Illien, Jens M. Turowski, Christoph Sens-Schonfelder, Clement Berenfeld, and Niels Hovius. This paper investigated the recovery rates of seismic velocity in near-surface materials after successive large earthquakes. The authors reported a correlation between the seismic velocities just before aftershocks and the subsequent rate of recovery, with faster recovery for drops with lower pre-drop velocity. Then they used a generic state variable theory to describe the superposition of multiple relaxations. Their theory provides a sound explanation for the differences in recovery rates after various aftershocks within earthquake sequences. I was very interested in their findings and enjoyed reading the manuscript. This paper is well written and clear. Hence, I support the publication of this article after corrections are made. Here are my specific comment on the theory that I hope will help to improve this interesting work as follows.*

AR: Thanks a lot for your enthusiasm in our paper, the summary in the first paragraph of the review really captures what we have done and the novelty for our work.

RC: *1. In my opinion, the state variable theory used in this manuscript is based on phenomena rather than physics. I would like to know if it is possible to predict recovery rates by simply superimposing the recovery processes of single earthquakes (as shown in figure below). If this idea works, the theory can be further simplified, and it is not necessary to average the state evolution for different τ .*

AR: Indeed, the presented state variable theory in our paper is based on phenomena as it would be challenging in the field to attribute the recovery dynamics to only one specific physical process.

On the schematics of your review, you define the same relaxation time τ_1 for event 1 and event 2, as the drop is initiated from the same δ_V baseline. This is correct and follows the interpretation in our paper. On the second plot of your review, you propose to superpose these two events, and we would get an apparent τ_2 . This is, in fact, what we do in Fig.2c of our paper (the red curve) and also this was our main hypothesis in an earlier study (Illien et al. [2022]) where we assumed a constant relaxation timescale for all seismic events. Our manuscript actually shows that the agreement with the Chile data using a constant timescale hypothesis (Figure 2c) is not as good as using the scaling showed in Figure 2b of the paper. We explained this scaling with the theory presented in the last part (especially on Figure 3c).

In both hypotheses (relaxation time is either constant, or controlled by the initial state of the subsurface just before the seismic event), we found remarkable that the recovery duration is actually more controlled by the subsurface state rather than the ground shaking amplitude. It is this finding we aim to emphasize in the manuscript.

Based on your review and the review 3, we have actually made a few adjustments to the text so that the modelling part is a bit clearer and ties better with our observations that are shown on Figures 1 and 2. Please

see the modifications made to the text under the modelling question of Reviewer 3 (the 8th comment).

Thanks a lot !

2. Reviewer 2

AR: We thank Reviewer 2 for their comments. We believe that some of the comments may be due to misunderstanding, while some other points are not aligned with published work. To reduce risk of misunderstanding, we have made some changes to the text.

2.1. Comments

RC: *1. This paper is about post-seismic changes in elastic wavespeed following earthquakes, therefore it will be of interest to those in seismology, geophysics and earthquake science, in addition to those interested in non-linear elasticity and transient creep. The paper would be of interest to a much broader group if it made connections to work done in Earth Science and in particular in earthquake science. Many previous works in those fields have documented changes in crustal elastic properties following earthquakes. Those studies show that seismic wave speed decreases abruptly following a mainshock and then recovers with log time. This is well known. I was surprised to see so little connection to these previous studies. The field observations are consistent with lab studies, expectations from rate/state friction theory, and studies of earthquake afterslip and fault healing. I was surprised to see so little connection to these previous studies.*

AR: We would like to comment on several points raised in this opening statement.

'The paper would be of interest to a much broader group if it made connections to work done in Earth Science and in particular in earthquake science' We feel that given the length of our paper (letter format), we already give justice to an important part of several Earth Science communities: Our introduction is motivated by observations from Seismologists (Brennguier et al. [2008], Gassenmeier et al. [2016], Viens et al. [2018]), Rock Physics experiments (TenCate et al. [2000], Shokouhi et al. [2017]) and Earth's Surface processes (Fan et al. [2019]), encompassing Geomorphology (Marc et al. [2016]) and Hydrology (Manga et al. [2012]). We also draw parallels with friction and contact dynamics literature (Dieterich [1979], Aharonov and Scholz [2019], Ostrovsky et al. [2019]) and civil engineering (Astorga et al. [2018]) in the last section of the paper, citing all these different communities when it's relevant in the text. For the 'Earthquake Science' comment, see our reply just under.

'The field observations are consistent with lab studies, expectations from rate/state friction theory, and studies of earthquake afterslip and fault healing. I was surprised to see so little connection to these previous studies.' We have made connections to the relevant literature with seminal work and the latest observations from all of these fields when they are connected to the statements in our paper. We mention the rate/state friction theory (Dieterich [1979]), contact dynamics friction frameworks Aharonov and Scholz [2019], Ostrovsky et al. [2019], crack closure physics in rocks deformation experiments Meyer et al. [2021] and slow dynamics studies (TenCate et al. [2000], Shokouhi et al. [2017])

For the earthquake afterslip and fault healing argument, we observe that most seismic interferometry studies conducted at frequencies above one Hertz suggest that fault properties have very little to do with the drop and recovery behaviour observed in dv/v . This is known due to poor correlation of drop amplitude and recovery in dv/v with co-seismic stress redistribution change computations Hobiger et al. [2014], Sawazaki et al. [2015], Wang et al. [2019]. This finding is not surprising as the waves would not sample the distant fault zone at high-frequency: in our study in Chile, we are located on a cliff at $\sim 100\text{km}$ from the subduction zone and outside the rupture area (Schurr et al. [2012, 2014]). It has been shown at this site already, that the strongest changes in dv/v induced by dynamic strain are visible at higher frequencies and at rather early time-windows in the correlation functions Richter et al. [2014], Gassenmeier et al. [2016], which shows that most of the physical changes due to the earthquakes are located towards the shallow subsurface. Moreover, a recent study by Steinmann et al. [2020] has shown that even small changes located very near the surface (5cm of frozen permafrost in their studies) can dominate the velocity changes measured at $> 1\text{Hz}$. Many of the seismic interferometry publications mentioned by reviewer 2 are from seismic deployment ON a fault zone (very

good work in California by Qin et al. [2020]) or performed with another method (Li [2017]) . This is less relevant to the work presented in this manuscript. We have added additional seismic interferometry literature but from contexts and methods that are comparable to our study:

Additional Seismic interferometry observations:

- Ermert et al. [2023]
- Okubo et al. [2024]

Now, we agree that this does not mean that relaxation/recovery is not active on fault interfaces at all, but in our study the recovery behaviour is observed in the subsurface materials far from the fault. Therefore, we cite the most appropriate literature with less emphasis on 'Earthquake fault' studies. To acknowledge possible parallels with fault interfaces, we have also added one sentence of implication for fault healing in our last paragraph:

For earthquake source physics and in particular for dynamic triggering of fault interfaces Gomberg and Johnson [2005], one could test if our findings apply e.g. the propensity to initiate aftershocks would be controlled by the interface properties during fault healing Bedford et al. [2023] rather by the dynamic strain amplitude.

'Those studies show that seismic wave speed decreases abruptly following a mainshock and then recovers with log time. This is well known.'The reviewer is correct, but we think that this comment is beside the point. We are very aware of the literature, and indeed there is nothing new in finding drop and recovery of seismic velocity (we point to this in the introduction:

It has been found that earthquakes induce a drop in subsurface seismic velocity, which is followed by a recovery on time scales ranging from a few days to several years Brenguier et al. [2008], Gassenmeier et al. [2016], Viens et al. [2018], Ermert et al. [2023], Okubo et al. [2024].

We actually use this widely reported observation as the main motivation for our study.

RC: 2. *There is quite a strong connection between this paper and the work of Gassenmeier et al. 2016. This paper needs some statements to clarify what's new here and the connections between these two papers*

AR: Indeed, our investigation used the same dataset than Gassenmeier et al. (2016) because of the unique suitability of station PATCX for our research question. We mention that in the introduction already:

Second, previous work has demonstrated that the station records allow highly stable measurements of very small velocity changes Richter et al. [2014], Gassenmeier et al. [2016], Sens-Schönfelder and Eulenfeld [2019], Illien et al. [2023].

However, our focus and analysis are totally new and different: **1.** We have refined the time-series with different processing and more data (Figure 1). **2.** We have analyzed all drop-recovery events and normalized them in a new way to highlight **the duration controls** of the relaxation phases (Figure 1 and 2). **3.** We build **an interpretation based on a general state variable theory** and highlighted how this theory should be modified when considering the numerous dv/v field data observations. None of these points are addressed in the Gassenmeier paper. Still, for highlighting the novelty of our work while acknowledging former studies at PATCX, we have written:

Using a 13-year time series of seismic velocity data for PATCX, we go beyond the former studies conducted at PATCX to show that the timescales of recovery can be predicted from pre-earthquake seismic velocity values regardless of the initial ground shaking intensity.

Otherwise, we believe that our goals are clearly stated in the introduction.

RC: *3. There are several statements about physical models for seismic velocity changes caused by earthquake shaking and about recovery. For example this sub-heading: ‘which physical framework can describe near-surface earthquake damage?’ This is an important goal. It would seem that this effort would start with or connect to the seminar work of O’Connell and Budiansky, 1974. But in fact, the model they use is simply a superposition of exponentials.*

AR: **’There are several statements about physical models for seismic velocity changes caused by earthquake shaking and about recovery.’**

We do not claim to build a physical model, but rather to capture the phenomenology, hence the expression ‘physical framework’. We have added the following clarification:

We used a generic state variable in a phenomenological model (Materials and Methods) that can characterise different processes, each with their own timescale (Fig. 3a).

Our goal was here to use a phenomenological approach that mimics the functional form of most physical relaxation frameworks. We chose this approach because all purely physical models are unconstrained in the field. One could fit a logarithmic recovery with a lot of different physical models and still wouldn’t know if their interpretation is the correct one... They do, however, have the same mathematical form (recovery following a logarithm), hence our approach. We chose to only take the analytical form of the models and show how it can be modified to replicate field observations and highlights future direction.

’It would seem that this effort would start with or connect to the seminar work of O’Connell and Budiansky, 1974.’ We agree that the work of O’Connell and Budiansky might be cited. However, the paper suggested by the reviewer is very general and doesn’t have the time-dependency aspect (it ‘only’ connects seismic velocities to the presence of flaws and cracks in a rock). Instead, we have added to the references another paper (which is still from Budiansky!), which was the first to connect seismic velocity recovery to crack closure to our knowledge:

- BUDIANSKY et al. [1982]

Finally, we have also added three more references from the rock physics literature:

- Schubnel et al. [2005]
- Brantut [2015]
- Geng et al. [2018]

’They used a simple superposition of exponentials.’ We think that the referee is mistaken, these are logarithmic recoveries. In figure S4b, we made the fit of our recoveries with the expression proposed by Snieder et al. [2016], which captures the logarithmic evolution of their universal relaxation function (equation 2 in our methods section). For our own phenomenological model, we show a proof that our state evolution obeys a logarithmic evolution (equation 10 in our method section). Second, the logarithmic function form we use corresponds to the empirical evolution of recovery time-series (as the referee observes in their first comment). To be sure, this is the solution form in all relaxation/slow dynamics models (BUDIANSKY et al. [1982], Aharonov and Scholz [2018], Ostrovsky et al. [2019])

RC: *4. Abstract: ... but its duration after earthquake ground shaking has not been constrained. This isn't true. Post-seismic phenomena such as afterslip and changes in wavespeed have been the subject of many studies. Many aspects of this problem are quite well understood. I agree that there are still interesting questions but this statement is misleading...*

AR: We disagree with the comment 'Many aspects of this problem are quite well understood'. Indeed, there's a lot of interesting research on the phenomena (as we have confirmed above). However, no attempt has been made yet to predict systematic behavior of the recovery duration in the field. We have refined our statement a little bit to be more precise:

systematic controls on the recovery duration in the shallow subsurface after earthquake ground shaking have not been determined

RC: *5. 'Abstract: We show that the relaxation time scale is a function of the state of the substrate at the time of seismic perturbation, rather than the intensity of ground shaking.' -> This is not true. The relaxation time scales directly with earthquake magnitude and ground shaking as measured by peak ground velocity (pgv).*

AR: We believe that this statement is not backed up by the literature, which is quite extensive on the subject.

Laboratory studies: The seminal work of TenCate et al. [2000] on slow dynamics and recovery already shows that the intensity of the perturbation has no impact on the duration of the recovery (Figure 5 of their paper showing the evolution of the resonance frequency following different perturbation intensity). This was confirmed recently by the work of Shokouhi et al. [2017], in which they measured the spectrum of relaxation timescales for different rocks and granular assemblies. They show that this spectrum is very consistent and does not change with the amplitude of the perturbation. So, in contrast to what the review 2 appears to claim. We could also cite other papers from the rock mechanics community but they are not as comparable, because they are at higher confining pressure in general. However, they also go in the same direction: in the recent study of Brantut [2015], we see that different axial strain amplitudes have very similar recovery duration (Figure 8 of his paper), even at high confining pressure.

Field Studies: First, the statement of the reviewer is directly contradicting the data of our current study. Moreover, no field studies have investigated thoroughly a PGV/PGA vs relaxation timescale relationship to our knowledge except one plot in Viens et al. [2018] for the Tohoku Earthquake. The R value in their scaling is 0.21 showing significant dispersion (Figure 8 of their paper). However, this study investigated the response to one earthquake at numerous sites, so we cannot exclude a dependence to site effects that would drive a spurious scaling between PGV and recovery timescales. We can also cite our earlier study in Nepal (Illien et al. [2022]), in which we showed a poor dependence of PGV value with relaxation duration.

One paper has tried to check the seismic velocity recovery rate depending on tectonic stresses (Pei et al. [2019]) but this was done with a different technique (using Pg waves sampling the entire crust), at lower frequency (10–25 s Rayleigh waves!) and with considerably fewer data points (Figure 2 of their paper). Their results suggest a possible effect due to stress but they sample very different depths from us.

Overall, did the referee perhaps mean to point at PGV/PGA scaling with velocity drops amplitude: such a scaling exists as shown in Richter et al. [2014], Gassenmeier et al. [2016], Viens et al. [2018], Illien et al. [2022]. However, this is not the topic of our paper.

RC: *6. Building upon the observation of universal relaxation in the lab, our study shows that after an earthquake, the mechanical properties of a subsurface medium control its relaxation behaviour and timescale of recovery, while the ground shaking intensity only controls the amplitude of the velocity change or the magnitude of damage. Following on this point, I don't see how you can make this claim. The timescale for recovery is directly related to the magnitude of the coseismic change in velocity. Larger magnitude changes require longer recovery times. This is clear in your Figures 1 and 2*

AR: We are not sure how the reviewer arrives to this conclusion from our figures (and the two other reviewers

understood well our message...). From our Figure 1, it is very apparent that Tocopilla and Iquique induced a different magnitude of coseismic velocity change (explained by different magnitudes/PGV), and yet they have very similar time scales of relaxation as shown on Figure 1a and even highlighted more by the normalisation on Figure 1d. To our eyes, this is quite clear... On Figure 2, we normalised all recoveries by the amplitude of the drop to check the recovery rate as a function of the pre-drop velocity. We believe that these figures speak for themselves, and against the statement of the referee.

3. Reviewer 3

Thanks a lot for your in-depth comments and interest in our study. We are certain that our manuscript benefits from the clarifications you requested.

3.1. Comments

RC: *1. Seismic interferometry: What is the depth sensitivity of waves within the frequency band used in this study? In the high-frequency band, body waves are expected to contribute to the noise correlations. In this case, the depth may exceed the shallow structure you are targeting. Is it possible to discuss changes in the near-surface materials even if body waves are assumed to dominate in the noise correlations?*

AR: Important comment. In the time-window we use for our correlation (7-16sec), we expect that the coda waves we used for the stretching analysis are scattered body-waves. In an earlier study (Illien et al. [2023]), we have estimated that a low velocity layer ($\sim 200\text{m}$) is present at our field site, which would focus the wave field in this layer. If we consider scattered wave sensitivity kernels (Pacheco and Snieder [2005], Obermann et al. [2013]), we would have a sensitivity that exponentially increase towards the surface. The study of Steinmann et al. [2020] shows that a few centimeter of frozen material at the subsurface can dominate the velocity changes measured above 1 Hz. We can also make the assumption of pure Rayleigh Waves (this would be unlikely given the time-window used in the correlation functions): at 3Hz (the lower band in our study) and at 1500 m/s velocity in the subsurface (Illien et al. [2023]), one wavelength would be around 500m (this is a crude estimation). A last argument would be that we would expect that the strongest physical changes occur at shallower depth.

In the end, we have hesitated with the terminology in the title: we could have replaced the term 'near-surface materials' with 'shallow subsurface' but we think that the latter is more vague as shallow subsurface means very different depths depending on who you ask (seismologist, rock physicists etc...). This, in the end, is an editorial matter, for discussion if a decision for publication is taken.

RC: *2. Temperature-correction: Please clarify which observation period was used for scaling temperature-induced dv/v . What was the scaling value estimated to be? And is that value reasonable compared to previous studies?*

AR: We have used an annual stack of the entirety of the time-series. For each annual time-series we have detrended the dv/v time-series to minimise the effect of earthquakes. In the year prior to the Tocopilla earthquake (2007), our correction yields very good agreement (the residuals are really flat on Figure S3 before Tocopilla). For the scaling value, we found the value of 0.05. However, this is hard to compare with the literature as the seismic processing and the temperature correction may differ between studies. The study of Richter et al. [2014] was at the same site than us but they used a full physical approach. However, their scaling can be guessed from the Figure 5 of their paper to be around ~ 0.025 , which is close to our value given the slight difference in seismic processing (3-6Hz, 7-16sec our study and 4-6Hz, 5-10s for Richter). There is also the recent study of Okubo et al. [2024], who find a value of $\sim 0.006-0.008$, but the comparison is challenging because this will be a function of the frequency bands used (0.9-1.2Hz), the time window in the correlation function (5s to 40s) and different site-specific conditions in California. We have added our inferred value in the manuscript for comparison:

We extracted the temperature at depth $z = 1.25$ m and scale it to the velocity changes through a factor α , which represent the expansion coefficient of the rocks in the subsurface ($\alpha \sim 0.053$).

RC: *3. The temperature model for paper is described as the average between two modeled dv/v . Is this averaging of different models appropriate? If the scaling factor (α) changes with time, how does this affect the temperature correction? I think it is possible that the sensitivity of dv/v to the temperature changes due to*

the large earthquakes.

AR: We have thought long and hard about this averaging problem. On one hand, we used an approach entirely based on the data (the sinusoidal model in Figure S1 and S3 in the paper), the other approach being more physical with the temperature. The drawback of the latter approach is that we have to rely on the temperature time-series measure at Iquique and not at exactly our field site (more discussion on that later in the response). Ultimately, we don't have enough constraints to choose the best approach and choose to take the average of both and benefit from the potential advantage of both methods.

We have calibrated the scaling factor based on a stack that encompass the entire time period to also average the impact of earthquake on thermal properties, therefore we do not resolve this effect. It is interesting to note that this question would be interesting to study in the future: The effect of temperature and earthquake damage is always sum up in a linear fashion, which is a simplification that is done in most of seismic interferometry paper Hobiger et al. [2014], Gassenmeier et al. [2016], Lecocq et al. [2017], Ermert et al. [2023], Okubo et al. [2024]. We found no significant differences when analyzing the response to earthquake during the warm or cold season. Reason for it may be due that temperature effects are mostly concentrated in the first few meters maximum while the effect of the earthquake are in general observed at a larger scale, across more frequency bands and time-windows of the cross-correlations (Richter et al. [2014] made this analysis at the PATCX field site). It is possible though that the temperature of the subsurface amplify or decrease ground motion by a few percent (Fokker et al. [2024]), which in turn could slightly influence the amplitude of the drop but we wouldn't expect to resolve this higher-order effect.

RC: **4. Figure 1a and state variable theory: In this study, a state variable model is used with $dv/dt=0$ as the recovery state. However, looking at Figure 1a, the seismic velocity change seems to be larger than the pre-drop value (e.g., the velocity from 2012 to 2014 after the Tocopilla earthquake). How should this be interpreted?**

AR: We haven't addressed the effect of a potential post-EQ new velocity level in our model. We have added a sentence to clarify that:

We look for an equation that is homogeneous to the velocity changes and choose a form that is mathematically viable with the logarithmic evolution of the recovery. We assume full recovery to the pre-earthquake seismic velocity state. A general equation for the velocity variations dv/dt can be expressed as

For how it should be interpreted, see the response to the next comment (very similar to this one :))

RC: **Figure 2c: An offset is visible between the observed velocity change and the model. What evidence supports attributing this offset to temperature correction? Please clarify the concept of a "new persistent velocity level." How does this relate to the statement that dv/v after the Tocopilla earthquake recovered to the pre-drop level?**

AR: Good question. We have attributed this offset to the temperature correction but also other possible source of uncertainties are possible as mentioned in the text.

For the temperature, the main source of uncertainty would be that the temperature dataset used in our model is taken at ~ 30 km from our field site. Although the airport of Atacama is in the same arid conditions, we may not have captured all temperature variations at PATCX. Another reason could be that the field site after Tocopilla recovers to a new higher velocity baseline, which would mean that the subsurface 'compacted' somehow. Some other studies, mention a new baseline but most of the time towards a lower velocity (permanent drop or damage), which would probably make more sense mechanically. An other reason that is never explored: following the Tocopilla earthquake, some scatterers may have change location in the subsurface, yielding small differences in the correlation function and this could lead to artefacts. Finally, we cannot exclude the potential effect of the long-term recovery of a potential earthquake that could have happen

anterior to our time-series.

In all these cases, this induce some uncertainty and variations in our data. This is why we have focus our analysis for the first 1000 days of recovery, where the seismic velocity variations are the strongest compared to the noise and offset in our data. We also wanted to minimise the number of parameters we had and present the simplest theory we have to avoid overfitting.

We have removed the 'persistent' term in the sentence for clarity:

The remaining offsets between the data and the synthetic time-series (particularly visible in the 2012-2014 period) may be due to the limitation of the temperature correction, a new ~~persistent~~ seismic velocity base level following the Tocopilla earthquake

RC: *5. Page4 ", all contacts are at steady state and the system does not evolve.": What specific situation is referred to when stating the system does not evolve? Does this imply that the number and orientation of cracks remain constant?*

AR: This is exact. If we take the example of cracks, the effective seismic velocity being very sensible to contact points along the crack surface (Kachanov and Sevostianov [2005]): an absence of seismic velocity variations would mean that their density and shape would not evolve anymore. More generally, this does not mean that 'nothing moves' at the microscopic level but in a statistical sense, the number of contacts stays the same: either nothing changes or the number of broken contacts equal the creation of others. We have re-framed a bit this sentence for clarity:

This combined contribution of different states of evolution is key to reconciling observations and physics. At the reference velocity value (in our study around 1500 m/s), ~~all contacts are at steady state~~ the density of contacts in the rocks is constant and the system ~~does not evolve~~ is at steady state.

RC: *6. How might the generation of new contacts or flaws by ground shaking, or changes in seismic structure anisotropy due to large earthquakes, affect velocity changes? These factors appear to be excluded from the current model.*

AR: This is an interesting point. Because we averaged all dv/v results for all combinations of the cross-correlations, we removed the influence of anisotropy. At our field site, the subsurface constitutes of clasts embedded in a matrix containing evaporites, likely gypsum and halite (Sens-Schönfelder et al. [2019]), forming a rather isotropic material. The influence of structural properties of a heterogeneous material on its seismic velocities occurs dominantly through changes of the contact area in the bond system between grains. In large scale observations an average over a large ensemble of contacts/cracks is intrinsically taken.

In general, there is not so much research that looked at monitoring anisotropic response to earthquake ground shaking with seismic interferometry. We agree that could be a good research question, especially in steep topographic terrain where we would expect anisotropy in the fracture network. Unfortunately, our field site is not optimised for that.

RC: *7. Figure S5: Please explain the term "phase portrait" in this context. While the figure shows different dv/dt values for the same $v(t)$, it is unclear how this reconciles the state variable argument with observations.*

AR: We agree that the text was not clear. We have made the following changes to the main text and to the legend of the figure:

Most existing physical and conceptual models associate the relaxation dynamics in seismic velocity in proportion to a 'state variable' representing the dynamics of the dominant type of contact or void embedded in a medium undergoing deformation. In these frameworks, the evolution of the variable of ~~of~~ inter-

est, here dv/dt , ~~is a direct function corresponds to a single value~~ of the variable itself ($v(t)$) ~~through a recovery function~~. This is the case in the rate and state friction framework Dieterich [1979] and models that involve contact dynamics Aharonov and Scholz [2018] and creeping mechanisms to fit relaxation data ~~Ostrovsky et al. [2019], Wang et al. [2021]~~ Geng et al. [2018], ~~Ostrovsky et al. [2019], Wang et al. [2021]~~. Field studies often ~~test-fit~~ such models based on one relaxation event. Our dataset, comprising 18 consecutive drop-and-recovery cycles does not support this state variable approach. ~~Importantly,~~ the phase portrait ~~of the relationship plot~~ between the rate of velocity change dv/dt and the velocity $v(t)$ shows very different phase locations after each seismic event (Fig. S5) i.e., the same ~~state velocity $v(t)$ can have different~~ rates of variation dv/dt ~~can be observed with different state velocity $v(t)$~~ . This indicates that the parameters of the fitted models would need to be changed at each drop and recovery cycle. Still, we suggest ~~that a state variable argument~~ how classic state variable functions can be reconciled with our observations ~~in the following~~.

RC: 8. Figure 3a: The determining v^* , v_0 , and v_{inf} when plotting this figure is unclear. A more detailed explanation how you created the figures would enhance reader understanding.

AR: This is true, sorry about that. We have now added a section at the end of the state variable theory to explain in plain language how we built the plot:

Using this framework, we computed the synthetic relative seismic velocity variations shown in Figure ??, which span a period of 10 000 days (~ 27 years). First, we arbitrarily define a logarithmic range of relaxation times τ from 10 to 4999 days (all values in Figure ??). For each of these τ values, the goal is to estimate a time-series $v_\tau(t)$ that would correspond to a relaxation phenomenon defined by its timescale τ . We fix v_∞ to 0 to obtain the amplitude of the velocity perturbations ($v_\tau(t)$ to $\Delta v_\tau(t)$) and defined 5 consecutive velocity drops Δ (-8, -2, -1, -1 and -5 m/s at respectively 500, 620, 1300, 5000, 7000 days in the time-series). Finally, we fix the value of v^* to 1000 m/s for all $\Delta v_\tau(t)$. Using these values, equation (??) can be numerically integrated to obtain the time-series shown in Figure ??a.

In a last step, we averaged all $\Delta v_\tau(t)$ perturbations following equation ?? and added the value $v_\infty = 1500$ m/s (reference value at our field site) to the time-series to obtain $v(t)$ as plotted on Figure ??b.

3.2. Additional comments

RC: Caption of Figure 2: Please consider using a center dot instead of asterisk to denote multiplication. Also, I think you should consider how many significant digits are needed.

AR: Thank you, done. We have reduced the number of digits. Because of the exponential dependence, the expression is very sensitive to the value of a and b .

The black lines show the best fit exponential scaling ($\tau_{max} = \exp(a * V_0 + b)$, $\tau_{max} = \exp(a \cdot V_0 + b)$ for $a = 5.22256 * 10^5.2226 \cdot 10^{-1}$ and $b = -7.764515 * 10^7.7645 \cdot 10^2$) and its standard deviation.

RC: Temperature-correction: As many readers may be unfamiliar with the Savitzky-Golay filter, please briefly explain its purpose and advantages over a simple moving average.

AR: We have added the following statement in the paper:

Temperature data were first smoothed with a Savitzky–Golay filter of 30 days length and order 3. The Savitzky-Golay filter smooths the data using a convolution with polynomials instead of a boxcar function (a simple moving average is a Savitzky-Golay filter of order 1). This has the advantage of retaining more high-frequency information in the time-series preserving more potential data peaks in

the smoothing window Schafer [2011].

RC: *"of of interest" -> "of interest"*

AR: Thanks, corrected.

References

- E. Aharonov and C. H. Scholz. A Physics-Based Rock Friction Constitutive Law: Steady State Friction. *Journal of Geophysical Research: Solid Earth*, 123(2):1591–1614, 2018. ISSN 21699356. .
- E. Aharonov and C. H. Scholz. The Brittle-Ductile Transition Predicted by a Physics-Based Friction Law. *Journal of Geophysical Research: Solid Earth*, 124(3):2721–2737, 2019. ISSN 21699356. .
- A. Astorga, P. Guéguen, and T. Kashima. Nonlinear elasticity observed in buildings during a long sequence of earthquakes. *Bulletin of the Seismological Society of America*, 108(3):1185–1198, 2018. ISSN 19433573. .
- J. D. Bedford, T. Hirose, and Y. Hamada. Rapid Fault Healing After Seismic Slip. *Journal of Geophysical Research: Solid Earth*, 128(6):e2023JB026706, 2023. . URL <https://agupubs.onlinelibrary.wiley.com/doi/abs/10.1029/2023JB026706>.
- N. Brantut. Time-dependent recovery of microcrack damage and seismic wave speeds in deformed limestone. *Journal of Geophysical Research: Solid Earth*, 120(12):8088–8109, 2015. .
- F. Brenguier, M. Campillo, C. Hadziioannou, N. M. Shapiro, R. M. Nadeau, and E. Larose. Postseismic relaxation along the San Andreas fault at Parkfield from continuous seismological observations. *Science (New York, N.Y.)*, 321(5895):1478–81, sep 2008. ISSN 1095-9203. .
- B. BUDIANSKY, J. W. HUTCHINSON, and S. SLUTSKY. Void Growth and Collapse in Viscous Solids. In H. G. HOPKINS and M. J. SEWELL, editors, *Mechanics of Solids*, pages 13–45. Pergamon, Oxford, 1982. ISBN 978-0-08-025443-2. . URL <https://www.sciencedirect.com/science/article/pii/B9780080254432500094>.
- J. H. Dieterich. Modeling of rock friction: 1. Experimental results and constitutive equations. *Journal of Geophysical Research: Solid Earth*, 84(B5):2161–2168, 1979. .
- L. A. Ermert, E. Cabral-Cano, E. Chaussard, D. Solano-Rojas, L. Quintanar, D. Morales Padilla, E. A. Fernández-Torres, and M. A. Denolle. Probing environmental and tectonic changes underneath Mexico City with the urban seismic field. *Solid Earth*, 14(5):529–549, 2023. . URL <https://se.copernicus.org/articles/14/529/2023/>.
- Y. Fan, M. Clark, D. M. Lawrence, S. Swenson, L. E. Band, S. L. Brantley, P. D. Brooks, W. E. Dietrich, A. Flores, G. Grant, J. W. Kirchner, D. S. Mackay, J. J. McDonnell, P. C. D. Milly, P. L. Sullivan, C. Tague, H. Ajami, N. Chaney, A. Hartmann, P. Hazenberg, J. McNamara, J. Pelletier, J. Perket, E. Rouholahnejad-Freund, T. Wagener, X. Zeng, E. Beighley, J. Buzan, M. Huang, B. Livneh, B. P. Mohanty, B. Nijssen, M. Safeeq, C. Shen, W. Verseveld, J. Volk, and D. Yamazaki. Hillslope Hydrology in Global Change Research and Earth System Modeling. *Water Resources Research*, 55(2):1737–1772, feb 2019. ISSN 0043-1397. .
- E. Fokker, E. Ruigrok, and J. Trampert. On the temperature sensitivity of near-surface seismic wave speeds: Application to the groningen region, the netherlands. *Geophysical Journal International*, 237, 03 2024. .
- M. Gassenmeier, C. Sens-Schönfelder, T. Eulendorf, M. Bartsch, P. Victor, F. Tilmann, and M. Korn. Field observations of seismic velocity changes caused by shaking-induced damage and healing due to mesoscopic nonlinearity. *Geophysical Journal International*, 204(3):1490–1502, 2016. ISSN 1365246X. .
- Z. Geng, A. Bonnelye, M. Chen, Y. Jin, P. Dick, C. David, X. Fang, and A. Schubnel. Time and Temperature Dependent Creep in Tournemire Shale. *Journal of Geophysical Research: Solid Earth*, 123(11):9658–9675, 2018. ISSN 21699356. .
- J. Gomberg and P. Johnson. Dynamic triggering of earthquakes. *Nature*, 437(7060):830, 2005. ISSN 1476-4687. . URL <https://doi.org/10.1038/437830a>.

- M. Hobiger, U. Wegler, K. Shiomi, and H. Nakahara. Single-station cross-correlation analysis of ambient seismic noise: application to stations in the surroundings of the 2008 Iwate-Miyagi Nairiku earthquake. *Geophysical Journal International*, 198(1):90–109, jul 2014. ISSN 1365-246X. .
- L. Illien, C. Sens-Schönfelder, C. Andermann, O. Marc, K. L. Cook, L. B. Adhikari, and N. Hovius. Seismic Velocity Recovery in the Subsurface: Transient Damage and Groundwater Drainage Following the 2015 Gorkha Earthquake, Nepal. *Journal of Geophysical Research: Solid Earth*, 127(2):1–18, 2022. ISSN 2169-9313. .
- L. Illien, C. Sens-Schönfelder, and K.-Y. Ke. Resolving minute temporal seismic velocity changes induced by earthquake damage: the more stations, the merrier? *Geophysical Journal International*, 234(1):124–135, 01 2023. ISSN 0956-540X. . URL <https://doi.org/10.1093/gji/ggad038>.
- M. Kachanov and I. Sevostianov. On quantitative characterization of microstructures and effective properties. In *International Journal of Solids and Structures*, volume 42, pages 309–336, jan 2005. .
- T. Lecocq, L. Longuevergne, H. A. Pedersen, F. Brenguier, and K. Stammler. Monitoring ground water storage at mesoscale using seismic noise: 30 years of continuous observation and thermo-elastic and hydrological modeling. *Scientific Reports*, 7(1), dec 2017. ISSN 20452322. .
- L. Li. Depth-dependence of post-seismic velocity changes in and near source area of the 2013 M7.0 Lushan earthquake revealed by S coda of repeating events. *Tectonophysics*, 717:302–310, 2017. ISSN 0040-1951. . URL <https://www.sciencedirect.com/science/article/pii/S0040195117303359>.
- M. Manga, I. Beresnev, E. E. Brodsky, J. E. Elkhoury, D. Elsworth, S. E. Ingebritsen, D. C. Mays, and C. Y. Wang. Changes in permeability caused by transient stresses: Field observations, experiments, and mechanisms. *Reviews of Geophysics*, 50(2), jun 2012. ISSN 87551209. .
- O. Marc, N. Hovius, P. Meunier, T. Gorum, and T. Uchida. A seismologically consistent expression for the total area and volume of earthquake-triggered landsliding. *Journal of Geophysical Research: Earth Surface*, 121(4):640–663, 2016. ISSN 21699011. .
- G. G. Meyer, N. Brantut, T. M. Mitchell, P. G. Meredith, and O. Plümper. Time Dependent Mechanical Crack Closure as a Potential Rapid Source of Post-Seismic Wave Speed Recovery: Insights From Experiments in Carrara Marble. *Journal of Geophysical Research: Solid Earth*, 126(4):1–29, 2021. ISSN 21699356. .
- A. Obermann, T. Planès, E. Larose, C. Sens-Schönfelder, and M. Campillo. Depth sensitivity of seismic coda waves to velocity perturbations in an elastic heterogeneous medium. *Geophysical Journal International*, 194(1):372–382, jul 2013. ISSN 0956540X. .
- K. Okubo, B. G. Delbridge, and M. A. Denolle. Monitoring Velocity Change Over 20 Years at Parkfield. *Journal of Geophysical Research: Solid Earth*, 129(4):1–38, 2024. ISSN 21699356. .
- L. Ostrovsky, A. Lebedev, J. Riviere, P. Shokouhi, C. Wu, M. A. Stuber Geesey, and P. A. Johnson. Long-Time Relaxation Induced by Dynamic Forcing in Geomaterials. *Journal of Geophysical Research: Solid Earth*, 124(5):5003–5013, 2019. ISSN 21699356. .
- C. Pacheco and R. Snieder. Time-lapse travel time change of multiply scattered acoustic waves. *The Journal of the Acoustical Society of America*, 118(3):1300–1310, sep 2005. ISSN 0001-4966. .
- S. Pei, F. Niu, Y. Ben-Zion, Q. Sun, Y. Liu, X. Xue, J. Su, and Z. Shao. Seismic velocity reduction and accelerated recovery due to earthquakes on the Longmenshan fault. *Nature Geoscience*, 12(5):387–392, may 2019. ISSN 17520908. .

- L. Qin, Y. Ben-Zion, L. F. Bonilla, and J. H. Steidl. Imaging and Monitoring Temporal Changes of Shallow Seismic Velocities at the Garner Valley Near Anza, California, Following the M7.2 2010 El Mayor-Cucapah Earthquake. *Journal of Geophysical Research: Solid Earth*, 125(1):1–17, 2020. ISSN 21699356. .
- T. Richter, C. Sens-Schönfelder, R. Kind, and G. Asch. Comprehensive observation and modeling of earthquake and temperature-related seismic velocity changes in northern Chile with passive image interferometry. *Journal of Geophysical Research: Solid Earth*, 119(6):4747–4765, 2014. ISSN 21699356. .
- K. Sawazaki, H. Kimura, K. Shiomi, N. Uchida, R. Takagi, and R. Snieder. Depth-dependence of seismic velocity change associated with the 2011 Tohoku earthquake, Japan, revealed from repeating earthquake analysis and finite-difference wave propagation simulation. *Geophysical Journal International*, 201(2): 741–763, 03 2015. ISSN 0956-540X. . URL <https://doi.org/10.1093/gji/ggv014>.
- R. W. Schafer. What Is a Savitzky-Golay Filter? [Lecture Notes]. (July):111–117, 2011.
- A. Schubnel, J. Fortin, L. Burlini, and Y. Guéguen. Damage and recovery of calcite rocks deformed in the cataclastic regime. *Geological Society Special Publication*, 245:203–221, 2005. ISSN 03058719. .
- B. Schurr, G. Asch, M. Rosenau, R. Wang, O. Oncken, S. Barrientos, P. Salazar, and J.-P. Vilotte. The 2007 M7.7 Tocopilla northern Chile earthquake sequence: Implications for along-strike and downdip rupture segmentation and megathrust frictional behavior. *Journal of Geophysical Research: Solid Earth*, 117 (B5), 2012. . URL <https://agupubs.onlinelibrary.wiley.com/doi/abs/10.1029/2011JB009030>.
- B. Schurr, G. Asch, S. Hainzl, J. Bedford, A. Hoechner, M. Palo, R. Wang, M. Moreno, M. Bartsch, Y. Zhang, O. Oncken, F. Tilmann, T. Dahm, P. Victor, S. Barrientos, and J. P. Vilotte. Gradual unlocking of plate boundary controlled initiation of the 2014 Iquique earthquake. *Nature*, 512(7514):299–302, 2014. ISSN 14764687. .
- C. Sens-Schönfelder and T. Eulenfeld. Probing the in situ Elastic Nonlinearity of Rocks with Earth Tides and Seismic Noise. *Physical Review Letters*, 122(13), apr 2019. ISSN 10797114. .
- C. Sens-Schönfelder, R. Snieder, and X. Li. A model for nonlinear elasticity in rocks based on friction of internal interfaces and contact aging. *Geophysical Journal International*, 216(1):319–331, jan 2019. ISSN 1365246X. .
- P. Shokouhi, J. Rivière, R. A. Guyer, and P. A. Johnson. Slow dynamics of consolidated granular systems: Multi-scale relaxation. *Applied Physics Letters*, 111(25), 2017. ISSN 00036951. .
- R. Snieder, C. Sens-Schönfelder, and R. Wu. The time dependence of rock healing as a universal relaxation process, a tutorial. *Geophysical Journal International*, 208(1):1–9, 2016. .
- R. Steinmann, C. Hadziioannou, and E. Larose. Effect of centimetric freezing of the near subsurface on Rayleigh and Love wave velocity in ambient seismic noise correlations. *Geophysical Journal International*, 224(1):626–636, 08 2020. ISSN 0956-540X. . URL <https://doi.org/10.1093/gji/ggaa406>.
- J. A. TenCate, E. Smith, and R. A. Guyer. Universal slow dynamics in granular solids. *Physical Review Letters*, 85(5):1020–1023, 2000. ISSN 00319007. .
- L. Viens, M. A. Denolle, N. Hirata, and S. Nakagawa. Complex Near-Surface Rheology Inferred From the Response of Greater Tokyo to Strong Ground Motions. *Journal of Geophysical Research: Solid Earth*, 123 (7):5710–5729, jul 2018. ISSN 21699356. .
- Q. Y. Wang, M. Campillo, F. Brenguier, A. Lecointre, T. Takeda, and A. Hashima. Evidence of Changes of Seismic Properties in the Entire Crust Beneath Japan After the Mw 9.0, 2011 Tohoku-oki Earthquake. *Journal of Geophysical Research: Solid Earth*, 124(8):8924–8941, 2019. ISSN 21699356. .

S. Y. Wang, H. Y. Zhuang, H. Zhang, H. J. He, W. P. Jiang, E. L. Yao, B. Ruan, Y. X. Wu, and Y. Miao. Near-surface softening and healing in eastern Honshu associated with the 2011 magnitude-9 Tohoku-Oki Earthquake. *Nature Communications*, 12(1):1–10, 2021. ISSN 20411723. .

Author's Response to Reviews of

Predictable recovery rates in near-surface materials after earthquake damage

Luc Illien, Jens Turowski, Christoph Sens-Schönfelder, Clement Berenfeld, Niels Hovius
Nature Communication,

RC: Reviewer's Comment, AR: Author's Response, Manuscript Text

1. Reviewer 1

Dear reviewer, thanks a lot for your feedback, we didn't understand properly your question about superposing the same maximum relaxation timescale in the first round. Your question was relevant and has led to some changes to the text and to the second figure. We are really happy because we feel the study has been augmented and is more elegant.

RC: *I read the author's response and the revised manuscript carefully. However, I still have some concerns on the necessity to average the state evolution for different τ .*

AR: Fundamentally, we actually need to consider the superposition of different timescales. The expression from Snieder et al. [2016], used to measure τ_{max} in our manuscript is the result of superposition of many relaxation timescales. Now, it is true the overall evolution of the system will be dominated by the longest relaxation timescale τ_{max} in the system. Following your comment, we found that the same τ_{max} can be applied after events (see our response after). However, we still need to consider the superposition of different state variables with their own timescales to explain this feature and point towards future physical models theories and relevant experiments.

RC: *I believe that the τ_{max} estimated by the interpolated average velocity recovery observed after Tocopilla and Iquique earthquakes (see Figure S4) does not reflect the relaxation timescale of seismic wave velocity after a single earthquake. Because the max derived from Figure S4 includes the influence of multiple aftershocks on the recovery process.*

AR: Indeed, we first fit the recovery, with the aftershocks as stated in the main text, which is likely to increase the estimated τ_{max} . Although, to the first order, it is not clear that the aftershocks induce a lot of perturbations as both recoveries after Tocopilla and Iquique are very similar. Nevertheless, we have now put this fitting in perspective in the text:

RC: *Aftershocks can cause the seismic wave velocity to take more time to recover to its pre-earthquake level, which means that in reality, the relaxation time of seismic wave velocity after a single earthquake will be much shorter than the fitted value of max. However, in Figure 2c of the manuscript, the authors used max as the relaxation time after a single earthquake to synthesize the recovery process, which is not very reasonable. I think using a smaller constant relaxation time may also yield a good result. It also should be noted that, as shown in Figure 2c, the seismic wave velocity can be restored to its pre-earthquake level in about 4 years after the Tocopilla and Iquique earthquakes. However, the max obtained in the manuscript is much longer than 4 years. From Figure 7 in Illien et al. (2012), it can be easily inferred that the max should be smaller than 4 years.*

AR: You are right. However, this figure from our 2022 paper is not the most appropriate for this argument. As you suggest in your schematics, we found (see new Figure 2 under) that the system can be properly modeled using the same τ_{max} value of 1.5-2 years after the main shocks and aftershocks, which leads to an overall

recovery after both events of ~ 3 years. We initially tried this hypothesis in our paper of 2022 but at the time, we couldn't fully explore the hypothesis because we had no velocity data from the period before the Gorkha earthquake. We note that this constant timescale of relaxation is also the source of the apparent scaling with the pre-drop velocities shown in Figure 2.

AR: We have made some adjustments to our discussion of the timescale in the manuscript. These changes reflect a sharpening of our principal argument and finding, and do not affect the conclusions of our work.

Abstract Changes:

Here, we analyse the effects of two successive large earthquakes and their aftershocks on ground properties using estimates of seismic velocity from ambient noise interferometry. We show that the relaxation time scale is a constant that is an intrinsic property of the substrate, independent of the intensity of ground shaking. Our study highlights the predictability of earthquake damage dynamics in the shallow subsurface and also in other materials.

Changes Introduction

And third, the hyper-arid setting precludes significant effects of changes in water content (~ 1 -5 mm/year of precipitation Houston [2006]). Using a 13-year time series of seismic velocity ~~data for PATCX changes~~, we go beyond the former studies conducted at PATCX to show that the ~~timescales of recovery subsurface response to multiple individual seismic events~~ can be predicted ~~from pre-earthquake seismic velocity values regardless of the initial ground shaking intensity~~ using a single value of recovery timescale (in this study $\sim 1.5 - 2$ years) independent of the various ground shaking intensities.

Changes Section 1: 13 years of seismic velocity changes induced by earthquake damage

Aftershocks with smaller attendant velocity drops did not have a significant impact on the total relaxation duration. We estimate that ~~on average~~, the full recovery of the seismic velocity after Tocopilla and Iquique earthquakes would take about ~~10 years (Materials and Methods, Fig. S4a)~~ 2.5-3 years (Fig. 1ad).

Changes Section 2: ~~Pre-drop velocity level controls the timescale of~~ A constant recovery timescale after seismic events

We observed that the velocity immediately prior to ground shaking, as well as the recovery timescales were very similar after the Tocopilla and Iquique earthquakes. This suggests that the subsurface relaxation is a function of the medium properties rather than the initial intensity of that ground shaking, and that the subsurface has an intrinsic recovery timescale. This hypothesis can be explored further by analysing the recoveries following aftershocks, extending the range of pre-drop velocity conditions prior to earthquakes. The first 10 days of recovery after each seismic event can be visualised and normalised by the pre-drop velocity level, here represented by the velocity on the day prior to the aftershock (Fig. 2a). A 10-day window was chosen to minimise the potential superposition of recoveries associated with previous aftershocks. In this time, recovery to the pre-aftershock baseline ranged from 20-30% for some aftershocks to 100% for others.

A correlation appears to exist between the seismic velocities just before aftershocks and the subsequent ~~rate of recovery~~ recovery duration, with faster recovery for drops with lower pre-drop velocity (Fig. 1a). We quantified this trend by fitting the 10-day recovery histories with a universal relaxation function Snieder et al. [2016] (Materials and Methods, Fig. S4b). This yields a likely maximum relaxation timescales τ_{\max} for each aftershock, where the value of τ_{\max} represents the recovery time to the seismic velocity observed before the drop. The distribution of the inferred timescales τ_{\max} for the aftershocks

in our time series ranges from ~ 8 to ~ 3000 days and can be described by an exponential relationship ($R^2 = 0.86$) with the pre-drop velocity. This scaling suggests that at low velocities relative to a baseline in which the velocities seem to be at steady state (here ~ 1500 m/s), the subsurface at PATCX may be more resilient with systematically shorter recovery times.

Using this fitted exponential relationship, we built a synthetic seismic velocity time-series. We measured all the velocity drops in the data, calculated their recovery timescale with their pre-drop velocity, and superposed them linearly, creating one time-series with multiple relaxations. Even if this synthetic time-series does not directly fit the full measured dataset, there is a good overall agreement with the instrumental data (Fig. 2c, Nash-Sutcliffe efficiency $NSE = 0.68$, Materials and Methods), ~~in contrast to a synthetic velocity time series calculated with~~. However, we can not rule out that the scaling of the pre-event velocity and the maximum relaxation timescale τ_{max} shown in Figure 2b may be due to the co-evolution of multiple recoveries characterized by a single constant recovery time scale after all earthquakes Illien et al. [2022]. In this scenario, the superposition of the same timescale of recovery following subsequent aftershocks could lead to an apparent scaling between the pre-drop velocity and the recovery timescale.

We test the hypothesis of a constant recovery time scale, as previously used Illien et al. [2022] (red line in after all seismic events using values for τ_{max} ranging from one to four years (Fig. 1e, $NSE = -2.61$), d). The ability of this method to predict the dv/v data outperforms the former approach with NSE values above 0.76 for $\tau_{max} \sim 1.5$ -2 years (Fig. 2e). For all models with constant τ_{max} , we also plot the recoveries normalised by the initial drop amplitudes to show the emergence of the apparent scaling between the pre-drop velocity and the recovery timescale (Fig. S5). This suggests that in the range of confining pressure and dynamic strain of our study, a single maximum recovery timescale τ_{max} value can be used to model the entire time-series. At PATCX, the total recovery duration observed after the Iquique and Tocopilla earthquakes is about three years but is in reality the result of multiple seismic events (the main shocks and their aftershocks) exciting a fundamental recovery timescale of 1.5-2 years. We note that a direct fit of the full average recovery after the main shocks (including their aftershocks sequences) would lead to a spurious duration τ_{max} of about 10 years (Fig. S4a).

The remaining offsets between the data and the ~~synthetic modeled~~ time-series (particularly visible in the 2012-2014 period) may be due to the limitation of the temperature correction, a new seismic velocity base level following the Tocopilla earthquake or a draw-down of groundwater levels in the Atacama region Viguier et al. [2019], which could affect the velocity changes Clements and Denolle [2018], Illien et al. [2021].

Changes Section 3: Which physical framework can describe near-surface earthquake damage ?

Our observations suggest that the timescale of seismic velocity recovery at a site is constant and can be predicted from the ~~known pre-drop velocity~~ analysis of former relaxations at a site and aftershock sequences. This feature is not directly tied to existing physical equations for relaxation. Most existing physical and conceptual models associate the relaxation dynamics in seismic velocity in proportion to a 'state variable' representing the dynamics of the dominant type of contact or void embedded in a medium undergoing deformation. In these frameworks, the evolution of the variable of interest, here dv/dt , corresponds to a single value of the variable itself $v(t)$ through a recovery function. This is the case in the rate and state friction framework Dieterich [1979] and models that involve contact dynamics Aharonov and Scholz [2018] and creeping mechanisms to fit relaxation data Geng et al. [2018], Ostrovsky et al. [2019], Wang et al. [2021]. Field studies often fit such models based on one relaxation event. Our dataset, comprising 18 consecutive drop-and-recovery cycles does not support this state variable approach: the phase portrait plot between the rate of velocity change dv/dt and the

velocity $v(t)$ shows very different phase locations after each seismic event (Fig. S6) i.e., the same rates of variation dv/dt can be observed with different state velocity $v(t)$. This indicates that the parameters of the fitted models would need to be changed at each drop and recovery cycle. Still, we suggest how classic state variable functions can be reconciled with our observations in the following.

In the field, seismic velocity measurements probe a subsurface volume that contains many types of contacts and flaws Snieder et al. [2016], Shokouhi et al. [2017], Sens-Schönfelder et al. [2019]. To account for their different dynamics, we assumed that all of these structures obey their own state variable equations, each with their timescale of relaxation. For instance, crack populations can be defined by their aspect ratios, which determine their time of closure under a specific load Budiansky et al. [1982], Meyer et al. [2021]. We used a generic state variable in a phenomenological model (Materials and Methods) that can ~~characterise~~ characterize different processes, each with their own timescale (Fig. 3a). The simple mean of all these state variable dynamics yields a synthetic time-series with the same features as our seismic velocity observations: A logarithmic recovery of velocity after each drop at a rate dependent on the pre-drop velocity (Fig. 2bc). Therefore, the superposition of different timescales in a sequence of perturbations can reproduce the dynamics we observed at station PATCX, which is dominated by the longest relaxation timescale in the system τ_{max} (Fig. 3). This combined contribution of different states of evolution is key to reconciling observations and physics. At the reference velocity value (in our study around 1500 m/s), the density of contacts in the rocks is constant and the system is at steady state. When the contacts are disturbed, they all enter a metastable state and recover at different rates. As the general reference velocity is approached, only the contact populations with the longest recovery times remain activated, while populations with a faster recovery have stabilized (Fig. 3). Assuming that the internal structures in the subsurface at a site are not fundamentally changed (no new types of contacts or flaws), we expect that the recovery of seismic velocity due to earthquakes can be predicted based on this theory and is the result of multiple constant timescales of relaxation. For higher state of damage (e.g after slip on fault interfaces), the timescales of recovery may change Kaproth and Marone [2014], Tinti et al. [2016], Rivière et al. [2016]. The boundary between these regimes is yet to be defined in future studies.

Building upon the observation of universal relaxation in the lab TenCate et al. [2000], our study shows that after an earthquake, the mechanical properties of a subsurface medium control its relaxation behaviour and timescale of recovery, while the ground shaking intensity only controls the amplitude of the velocity change or the magnitude of damage. If this recovery behaviour is valid for near-surface geomaterials, then it may also be tested for man-made structures Astorga et al. [2018]. Understanding the link between specific types of flaws and relaxation timescales can lead to a targeted material test for management of the response to damage. For earthquake source physics and in particular for dynamic triggering of fault interfaces Gomberg and Johnson [2005], one could test if our findings apply e.g. the propensity to initiate aftershocks would be controlled by the interface properties during fault healing Bedford et al. [2023] rather by the dynamic strain amplitude. A representative relaxation timescale may also allow an estimation of the propensity to failure of steep topographic slopes in an earthquake epicentral area Marc et al. [2021] and therefore an assessment of the feasibility and timeliness of reconstruction after large earthquakes. This can reduce lingering post-seismic risk, and minimize spurious investment. Our results should also be tested in presence of pronounced anisotropy in which different recovery duration could be observed with variable seismic polarizations. Finally, we propose that measuring the rate of velocity change over a period immediately after an episode of strong ground motion and before any damaging aftershocks, can yield early constraints on the likely pace of recovery of substrate damage. This early estimation can be based on measurements from small seismic events to ~~construct a site-dependent relationship between pre-drop seismic velocity and the expected relaxation timescale as shown in Fig. 1b,~~ constrain the likely maximum relaxation τ_{max} at a specific site and then upscaled for larger earthquakes and their aftershock sequences.

Figure 1: *Pre-drop velocity level as a control on the relaxation timescale and corresponding synthetic time-series.* **a.** Recoveries after individual seismic events, normalised by the amplitude of the drop. The colour indicates the relevant pre-drop velocity. The thick line shows the early average recovery observed after the To-copilla and Iquique main shocks. **b.** Fitted maximum recovery timescales for the recoveries shown in **a** ($N=18$). The black lines show the best fit exponential scaling ($\tau_{max} = \exp(a \cdot V_0 + b)$) and its standard deviation ($R^2=0.86, N=18$). **c.** Synthetic velocity time-series built by superposition of relaxations obtained with the relationship shown in **b** (blue) and for a shows the instrumental data (grey). **d.** Synthetic velocity time-series built by superposition of constant timescales of relaxation timescale (in red and taken as 3887 d, indicated by the value fitted for colour of the Iquique and To-copilla recoveries curves). **e.** Nash-Sutcliffe efficiency (NSE) for all tested models with constant τ_{max} ; the colours correspond to relaxation timescales shown in **d**. The grey dashed line shows the instrumental data NSE value for the dv/v model obtained with the scaling on **c**.

2. Reviewer 2

RC: *The authors have done good work to revise the manuscript. They have addressed the questions of all reviewers and the paper's message is now clearer. This is a nice piece of work. I have only one outstanding*

issue and a few minor points.

AR: Thanks a lot for your comments and response.

RC: *The main issues relates to this statement, found in the abstract and elsewhere: “We show that the relaxation time scale is a function of the state of the substrate at the time of seismic perturbation, rather than the intensity of ground shaking.” In fact, while I see their point but while it holds for the data shown in fig. 1 it cannot be true in an absolute sense for all states of shaking induced damage. Certainly, if the state of damage is sufficient recovery will take longer. This fact is also clear when one considers data on frictional strength recovery –frictional healing. In studies of healing one observes that the degree of strength recovery scales with time, and this applies not only to frictional strength but also to elastic properties including seismic wave speed (for example Ref 1 below). Here, one sees that the longer we wait the more the wave speed increases. I do not think that this contradicts their results, but rather adds to it and perhaps suggests a range of behaviors. Having read their rebuttal, I believe that our differences are not fundamental but rather a matter of degree. So, I would be satisfied if they used some qualification to these statements.*

AR: Thank you for your comment. We agree that our observations are valid for dynamic strain regimes and that the pictures may be different when approaching processes at higher confining pressure and/or closer to fault interfaces. We have made an amendment to specify further the specificity of our study

In the third section of the results/discussion:

Assuming that the internal structures in the subsurface at a site are not fundamentally changed (no new types of contacts or flaws), we expect that the recovery of seismic velocity due to earthquakes can be predicted based on this theory and is the result of multiple constant timescales of relaxation. For higher state of damage (e.g after slip on fault interfaces), the timescales of recovery may change Kaproth and Marone [2014], Tinti et al. [2016], Rivière et al. [2016]. The boundary between these regimes is yet to be defined in future studies.

In the second section of the results/discussion

This suggests that in the range of confining pressure and dynamic strain of our study, a single maximum recovery timescale τ_{\max} value can be used to model the entire time-series.

RC: *Other and minor points. A. Figure 1: Dots at the top mark the occurrence of major aftershocks.*

AR: Thank you, we have made the following change to the legend:

Dots at the top mark the ~~occurrence~~occurrence of major aftershocks.

RC: *B. Yes, I see that you have added references to geophysical observations. But for me, there are still too many statements that are unqualified or insufficiently tied to previous observations. For example, in the Introduction, this sentence should also include reference to a few of the studies below. “It has been found that earthquakes induce a drop in subsurface seismic velocity, which is followed by a recovery on time scales ranging from a few days to several years.” The references they have now (5-9) are valid but do not include basic studies of seismic damage and post-seismic recovery. I do not discount their references to 11-13, but there are important field studies that connect strongly to their work and should be included.*

AR: Thank you, we have now added all the studies you mention in your review.

In the introduction:

It has been found that earthquakes induce a drop in subsurface seismic velocity, which is followed by a recovery on time scales ranging from a few days to several years Brenquier et al. [2008], Gassenmeier et al. [2016], Viens et al. [2018], Brenquier et al. [2008], Li et al. [1998], Vidale and Li [2003], Cochran et al. [2009], Riviere et al. [2013], Viens et al. [2018], F

and in the sentence we have added about the dynamic strain regime (the quote is just above under your previous question).

RC: *Why is it called ‘pre-drop’ velocity? Wouldn’t initial velocity be better?*

AR: In an earlier version of the draft, we actually called it initial velocity but it confused some of our co-authors as it can be understood as the velocity at the beginning of the recovery. We found that pre-drop was the clearer way to pinpoint to the velocity just before the drop.

3. Reviewer 3

RC: *The authors politely responded to my previous reviews. I would like to ask an additional question regarding the model of velocity time series based on state variable theory. According to the authors' response, it seems that the effect of velocity anisotropy due to large earthquake does not need to be considered in this field. However, could individual large earthquakes generate new cracks and flaws? If so, might this introduce discrepancies between the model and the data presented in Figure 2?*

AR: Thank you for your comment. To be clear, we also believe that anisotropy would be important if one would want to measure the seismic velocity response using different polarizations. In our case, to increase the signal to noise ratio of dv/v , we have averaged all combinations, therefore averaging the potential anisotropic response from all directions. Another reason to simplify our approach is that the subsurface at PATCX is composed of a conglomerate with evaporites inside (Sens-Schönfelder and Eulenfeld [2019]): we would expect flaws to be distributed in many different directions in this type of rocks, with little anisotropy. In a future study, one may imagine different timescales spectrum depending on the wave propagation direction and incidence of the earthquakes. We have added a sentence of perspective on that in the last paragraph:

This can reduce lingering post-seismic risk, and minimize spurious investment. Our results should also be tested in presence of pronounced anisotropy in which different recovery duration could be observed with variable seismic polarizations.

We are also in the regime of dynamic strain and we assume that the material hasn't fundamentally changed its characteristic flaws/voids distribution (no new cracks and flaws, or no introduction of entropy from a thermodynamic point of view). This explains our assumption that the velocity comes back to pre-earthquake values (all changes are recovered) in our approach. We have precised that a bit more in the text.

In the method section about the state variable:

We assume full recovery to the pre-earthquake seismic velocity state (No generation of new contacts/flaws). A general equation for the velocity variations dv/dt can be expressed as

We also precise in the main text of the last section

Assuming that the internal structures in the subsurface at a site are not fundamentally changed (no new types of contacts or flaws), we expect that the recovery of seismic velocity due to earthquakes can be predicted based on this theory and is the result of multiple constant timescales of relaxation. For higher state of damage (e.g after slip on fault interfaces), the timescales of recovery may change Kaproth and Marone [2014], Tinti et al. [2016], Rivière et al. [2016]. The boundary between these regimes is yet to be defined in future studies.

RC: *Furthermore, this study seems to focus solely on the effects of dynamic stress. Could static stress also influence structural changes? The spatial distribution of static stress, whether expansive or contractive, likely varies for each earthquake at the observed location. Would this variability affect the recovery process of cracks or flaws?*

AR: Indeed, static changes at higher confining pressure induce relaxation as well (Brantut [2015]). However, previous analysis of seismic velocity changes at high-frequency (>1Hz) showed very little correlation with static stress redistribution (Hobiger et al. [2014], Sawazaki et al. [2015], Wang et al. [2019]). The potential reason is that seismic interferometry is more sensitive to shallow changes in the near-surface materials: in our case, the station is outside the coseismic slip zone of the earthquakes (Schurr et al. [2012, 2014]), therefore we attribute the changes to dynamic strain.

RC: *First line of page 2: seismic station station -> seismic station*

AR: Corrected

References

- E. Aharonov and C. H. Scholz. A Physics-Based Rock Friction Constitutive Law: Steady State Friction. *Journal of Geophysical Research: Solid Earth*, 123(2):1591–1614, 2018. ISSN 21699356. .
- A. Astorga, P. Guéguen, and T. Kashima. Nonlinear elasticity observed in buildings during a long sequence of earthquakes. *Bulletin of the Seismological Society of America*, 108(3):1185–1198, 2018. ISSN 19433573. .
- J. D. Bedford, T. Hirose, and Y. Hamada. Rapid Fault Healing After Seismic Slip. *Journal of Geophysical Research: Solid Earth*, 128(6):e2023JB026706, 2023. . URL <https://agupubs.onlinelibrary.wiley.com/doi/abs/10.1029/2023JB026706>.
- N. Brantut. Time-dependent recovery of microcrack damage and seismic wave speeds in deformed limestone. *Journal of Geophysical Research: Solid Earth*, 120(12):8088–8109, 2015. .
- F. Brenguier, M. Campillo, C. Hadziioannou, N. M. Shapiro, R. M. Nadeau, and E. Larose. Postseismic relaxation along the San Andreas fault at Parkfield from continuous seismological observations. *Science (New York, N.Y.)*, 321(5895):1478–81, sep 2008. ISSN 1095-9203. .
- B. Budiansky, J. W. Hutchinson, and S. Slutsky. Void Growth and Collapse in Viscous Solids. In H. G. Hopkins and M. J. Sewell, editors, *Mechanics of Solids*, pages 13–45. Pergamon, Oxford, 1982. ISBN 978-0-08-025443-2. . URL <https://www.sciencedirect.com/science/article/pii/B9780080254432500094>.
- T. Clements and M. A. Denolle. Tracking Groundwater Levels Using the Ambient Seismic Field. *Geophysical Research Letters*, 45(13):6459–6465, jul 2018. ISSN 19448007. .
- E. S. Cochran, Y.-G. Li, P. M. Shearer, S. Barbot, Y. Fialko, and J. E. Vidale. Seismic and geodetic evidence for extensive, long-lived fault damage zones. *Geology*, 37(4):315–318, 2009. ISSN 0091-7613. . URL <https://doi.org/10.1130/G25306A.1>.
- J. H. Dieterich. Modeling of rock friction: 1. Experimental results and constitutive equations. *Journal of Geophysical Research: Solid Earth*, 84(B5):2161–2168, 1979. .
- L. A. Ermert, E. Cabral-Cano, E. Chaussard, D. Solano-Rojas, L. Quintanar, D. Morales Padilla, E. A. Fernández-Torres, and M. A. Denolle. Probing environmental and tectonic changes underneath Mexico City with the urban seismic field. *Solid Earth*, 14(5):529–549, 2023. . URL <https://se.copernicus.org/articles/14/529/2023/>.
- M. Gassenmeier, C. Sens-Schönfelder, T. Eulenfeld, M. Bartsch, P. Victor, F. Tilmann, and M. Korn. Field observations of seismic velocity changes caused by shaking-induced damage and healing due to mesoscopic nonlinearity. *Geophysical Journal International*, 204(3):1490–1502, 2016. ISSN 1365246X. .
- Z. Geng, A. Bonnelye, M. Chen, Y. Jin, P. Dick, C. David, X. Fang, and A. Schubnel. Time and Temperature Dependent Creep in Tournemire Shale. *Journal of Geophysical Research: Solid Earth*, 123(11):9658–9675, 2018. ISSN 21699356. .
- J. Gomberg and P. Johnson. Dynamic triggering of earthquakes. *Nature*, 437(7060):830, 2005. ISSN 1476-4687. . URL <https://doi.org/10.1038/437830a>.
- M. Hobiger, U. Wegler, K. Shiomi, and H. Nakahara. Single-station cross-correlation analysis of ambient seismic noise: application to stations in the surroundings of the 2008 Iwate-Miyagi Nairiku earthquake. *Geophysical Journal International*, 198(1):90–109, jul 2014. ISSN 1365-246X. .
- J. Houston. Variability of precipitation in the Atacama Desert: Its causes and hydrological impact. *International Journal of Climatology*, 26(15):2181–2198, 2006. ISSN 08998418. .

- L. Illien, C. Andermann, C. Sens-Schönfelder, K. L. Cook, K. P. Baidya, L. B. Adhikari, and N. Hovius. Subsurface Moisture Regulates Himalayan Groundwater Storage and Discharge. *AGU Advances*, 2(2), 2021. ISSN 2576-604X. .
- L. Illien, C. Sens-Schönfelder, C. Andermann, O. Marc, K. L. Cook, L. B. Adhikari, and N. Hovius. Seismic Velocity Recovery in the Subsurface: Transient Damage and Groundwater Drainage Following the 2015 Gorkha Earthquake, Nepal. *Journal of Geophysical Research: Solid Earth*, 127(2):1–18, 2022. ISSN 2169-9313. .
- B. M. Kaproth and C. Marone. Evolution of elastic wave speed during shear-induced damage and healing within laboratory fault zones. *Journal of Geophysical Research: Solid Earth*, 119(6):4821–4840, 2014. . URL <https://agupubs.onlinelibrary.wiley.com/doi/abs/10.1002/2014JB011051>.
- Y.-G. Li, J. E. Vidale, K. Aki, F. Xu, and T. Burdette. Evidence of Shallow Fault Zone Strengthening After the 1992 *M*_{7.5} Landers, California, Earthquake. *Science*, 279(5348):217–219, 1998. . URL <https://www.science.org/doi/abs/10.1126/science.279.5348.217>.
- O. Marc, C. Sens-Schönfelder, L. Illien, P. Meunier, M. Hobiger, K. Sawazaki, C. Rault, and N. Hovius. Toward Using Seismic Interferometry to Quantify Landscape Mechanical Variations after Earthquakes. *Bulletin of the Seismological Society of America*, pages 1–19, 2021. ISSN 0037-1106. .
- G. G. Meyer, N. Brantut, T. M. Mitchell, P. G. Meredith, and O. Plümper. Time Dependent Mechanical Crack Closure as a Potential Rapid Source of Post-Seismic Wave Speed Recovery: Insights From Experiments in Carrara Marble. *Journal of Geophysical Research: Solid Earth*, 126(4):1–29, 2021. ISSN 21699356. .
- K. Okubo, B. G. Delbridge, and M. A. Denolle. Monitoring Velocity Change Over 20 Years at Parkfield. *Journal of Geophysical Research: Solid Earth*, 129(4):1–38, 2024. ISSN 21699356. .
- L. Ostrovsky, A. Lebedev, J. Riviere, P. Shokouhi, C. Wu, M. A. Stuber Geesey, and P. A. Johnson. Long-Time Relaxation Induced by Dynamic Forcing in Geomaterials. *Journal of Geophysical Research: Solid Earth*, 124(5):5003–5013, 2019. ISSN 21699356. .
- J. Riviere, T. Candela, M. Scuderi, C. Marone, R. Guyer, and P. A. Johnson. Dynamic acousto-elasticity in Berea sandstone: Influence of the strain rate. *The Journal of the Acoustical Society of America*, 134(5*supplement*): 4197, 2013. ISSN 0001 – 4966. .URL.
- J. Rivière, L. Pimienta, M. Scuderi, T. Candela, P. Shokouhi, J. Fortin, A. Schubnel, C. Marone, and P. A. Johnson. Frequency, pressure, and strain dependence of nonlinear elasticity in Berea Sandstone. *Geophysical Research Letters*, 43(7):3226–3236, 2016. . URL <https://agupubs.onlinelibrary.wiley.com/doi/abs/10.1002/2016GL068061>.
- K. Sawazaki, H. Kimura, K. Shiomi, N. Uchida, R. Takagi, and R. Snieder. Depth-dependence of seismic velocity change associated with the 2011 Tohoku earthquake, Japan, revealed from repeating earthquake analysis and finite-difference wave propagation simulation. *Geophysical Journal International*, 201(2): 741–763, 03 2015. ISSN 0956-540X. . URL <https://doi.org/10.1093/gji/ggv014>.
- B. Schurr, G. Asch, M. Rosenau, R. Wang, O. Oncken, S. Barrientos, P. Salazar, and J.-P. Vilotte. The 2007 M7.7 Tocopilla northern Chile earthquake sequence: Implications for along-strike and downdip rupture segmentation and megathrust frictional behavior. *Journal of Geophysical Research: Solid Earth*, 117 (B5), 2012. . URL <https://agupubs.onlinelibrary.wiley.com/doi/abs/10.1029/2011JB009030>.

- B. Schurr, G. Asch, S. Hainzl, J. Bedford, A. Hoechner, M. Palo, R. Wang, M. Moreno, M. Bartsch, Y. Zhang, O. Oncken, F. Tilmann, T. Dahm, P. Victor, S. Barrientos, and J. P. Vilotte. Gradual unlocking of plate boundary controlled initiation of the 2014 Iquique earthquake. *Nature*, 512(7514):299–302, 2014. ISSN 14764687. .
- C. Sens-Schönfelder and T. Eulenfeld. Probing the in situ Elastic Nonlinearity of Rocks with Earth Tides and Seismic Noise. *Physical Review Letters*, 122(13), apr 2019. ISSN 10797114. .
- C. Sens-Schönfelder, R. Snieder, and X. Li. A model for nonlinear elasticity in rocks based on friction of internal interfaces and contact aging. *Geophysical Journal International*, 216(1):319–331, jan 2019. ISSN 1365246X. .
- P. Shokouhi, J. Rivière, R. A. Guyer, and P. A. Johnson. Slow dynamics of consolidated granular systems: Multi-scale relaxation. *Applied Physics Letters*, 111(25), 2017. ISSN 00036951. .
- R. Snieder, C. Sens-Schönfelder, and R. Wu. The time dependence of rock healing as a universal relaxation process, a tutorial. *Geophysical Journal International*, 208(1):1–9, 2016. .
- J. A. TenCate, E. Smith, and R. A. Guyer. Universal slow dynamics in granular solids. *Physical Review Letters*, 85(5):1020–1023, 2000. ISSN 00319007. .
- E. Tinti, M. M. Scuderi, L. Scognamiglio, G. Di Stefano, C. Marone, and C. Collettini. On the evolution of elastic properties during laboratory stick-slip experiments spanning the transition from slow slip to dynamic rupture. *Journal of Geophysical Research: Solid Earth*, 121(12):8569–8594, 2016. . URL <https://agupubs.onlinelibrary.wiley.com/doi/abs/10.1002/2016JB013545>.
- J. E. Vidale and Y.-G. Li. Damage to the shallow Landers fault from the nearby Hector Mine earthquake. *Nature*, 421(6922):524–526, 2003. ISSN 1476-4687. . URL <https://doi.org/10.1038/nature01354>.
- L. Viens, M. A. Denolle, N. Hirata, and S. Nakagawa. Complex Near-Surface Rheology Inferred From the Response of Greater Tokyo to Strong Ground Motions. *Journal of Geophysical Research: Solid Earth*, 123(7):5710–5729, jul 2018. ISSN 21699356. .
- B. Viguié, H. Jourde, V. Leonardi, L. Daniele, C. Batiot-Guilhe, G. Favreau, and V. De Montety. Water table variations in the hyperarid Atacama Desert: Role of the increasing groundwater extraction in the pampa del tamarugal (Northern Chile). *Journal of Arid Environments*, 168:9–16, 2019. ISSN 0140-1963. . URL <https://www.sciencedirect.com/science/article/pii/S0140196318309790>.
- Q. Y. Wang, M. Campillo, F. Brenguier, A. Lecointre, T. Takeda, and A. Hashima. Evidence of Changes of Seismic Properties in the Entire Crust Beneath Japan After the Mw 9.0, 2011 Tohoku-oki Earthquake. *Journal of Geophysical Research: Solid Earth*, 124(8):8924–8941, 2019. ISSN 21699356. .
- S. Y. Wang, H. Y. Zhuang, H. Zhang, H. J. He, W. P. Jiang, E. L. Yao, B. Ruan, Y. X. Wu, and Y. Miao. Near-surface softening and healing in eastern Honshu associated with the 2011 magnitude-9 Tohoku-Oki Earthquake. *Nature Communications*, 12(1):1–10, 2021. ISSN 20411723. .

Author's Response to Reviews of

Predictable recovery rates in near-surface materials after earthquake damage

Luc Illien, Jens Turowski, Christoph Sens-Schönfelder, Clement Berenfeld, Niels Hovius
Nature Communication,

RC: Reviewer's Comment, AR: Author's Response, Manuscript Text

1. Reviewers feedback

RC: Reviewer 1 (Remarks to the Author):

I am very grateful to the authors for providing a detailed response to my comments. This paper demonstrates that stacking recovery processes with different recovery times can better predict the recovery process of near surface wave velocity. The error between observation and simulation (i.e. the Nash Sutcliff efficiency value mentioned in the article) decreased from 0.76 to 0.68. This result is very interesting and can deepen our understanding of the slow dynamics of geotechnical materials. Therefore, I agree to the publication of this paper.

RC: Reviewer 2 (Remarks to the Author):

The authors have now responded fully to my reviews and, in my opinion, to those of the other reviewers. This is a very nice paper and it is ready to be published.

RC: Reviewer 3 (Remarks to the Author):

The authors have addressed all my comments and I am happy to recommend publication of this article.

2. Author's response

AR: Thank you everyone for your feedback ! We are grateful for your reviews !

I have carefully reviewed the manuscript titled “Predictable recovery rates in near-surface materials after earthquake damage” by Luc Illien, Jens M. Turowski, Christoph Sens-Schonfelder, Clement Berenfeld, and Niels Hovius. This paper investigated the recovery rates of seismic velocity in near-surface materials after successive large earthquakes. The authors reported a correlation between the seismic velocities just before aftershocks and the subsequent rate of recovery, with faster recovery for drops with lower pre-drop velocity. Then they used a generic state variable theory to describe the superposition of multiple relaxations. Their theory provides a sound explanation for the differences in recovery rates after various aftershocks within earthquake sequences. I was very interested in their findings and enjoyed reading the manuscript. This paper is well written and clear. Hence, I support the publication of this article after corrections are made. Here are my specific comment on the theory that I hope will help to improve this interesting work as follows.

In my opinion, the state variable theory used in this manuscript is based on phenomena rather than physics. I would like to know if it is possible to predict recovery rates by simply superimposing the recovery processes of single earthquakes (as shown in figure below). If this idea works, the theory can be further simplified, and it is not necessary to average the state evolution for different τ .

[figure redacted]

Review

This is an interesting paper on an important topic. For the most part, it's well written and the science is sound. But there are quite a few problems, some major, that would need to be addressed before it could be considered for publication. These are noted below along with some references that should be included.

1. This paper is about post-seismic changes in elastic wavespeed following earthquakes, therefore it will be of interest to those in seismology, geophysics and earthquake science, in addition to those interested in non-linear elasticity and transient creep. The paper would be of interest to a much broader group if it made connections to work done in Earth Science and in particular in earthquake science. Many previous works in those fields have documented changes in crustal elastic properties following earthquakes. Those studies show that seismic wave speed decreases abruptly following a mainshock and then recovers with log time. This is well known. I was surprised to see so little connection to these previous studies. The field observations are consistent with lab studies, expectations from rate/state friction theory, and studies of earthquake afterslip and fault healing. I was surprised to see so little connection to these previous studies.
2. There is quite a strong connection between this paper and the work of Gassenmeier et al. 2016. This paper needs some statements to clarify what's new here and the connections between these two papers.
3. There are several statements about physical models for seismic velocity changes caused by earthquake shaking and about recovery. For example this sub-heading: 'which physical framework can describe near-surface earthquake damage?' This is an important goal. It would seem that this effort would start with or connect to the seminar work of O'Connell and Budiansky, 1974. But in fact, the model they use is simply a superposition of exponentials.
4. Abstract: *...but its duration after earthquake ground shaking has not been constrained.*
This isn't true. Post-seismic phenomena such as afterslip and changes in wavespeed have been the subject of many studies. Many aspects of this problem are quite well understood. I agree that there are still interesting questions but this statement is misleading...
5. Abstract: *We show that the relaxation time scale is a function of the state of the substrate at the time of seismic perturbation, rather than the intensity of ground shaking.*
This is not true. The relaxation time scales directly with earthquake magnitude and ground shaking as measured by peak ground velocity (pgv).
6. *Building upon the observation of universal relaxation in the lab, our study shows that after an earthquake, the mechanical properties of a subsurface medium control its relaxation behaviour and timescale of recovery, while the ground shaking intensity only controls the amplitude of the velocity change or the magnitude of damage.*

Following on this point, I don't see how you can make this claim. The timescale for recovery is directly related to the magnitude of the coseismic change in velocity. Larger magnitude changes require longer recovery times. This is clear in your Figures 1 and 2.

References

- Brenguier, F., Campillo, M., Hadziioannou, C., Shapiro, N. M., Nadeau, R. M., & Larose, E. (2008), Postseismic Relaxation Along the San Andreas Fault at Parkfield from Continuous Seismological Observations., *Science*
- E. J. Chaves, S. Y. Schwartz, and R. E. Abercrombie, Repeating earthquakes record fault weakening and healing in areas of megathrust postseismic slip, 2020.
- Li, L., Depth-dependence of post-seismic velocity changes in and near source area of the 2013 M7.0 Lushan earthquake revealed by S coda of repeating events, *Tectonophysics*, 2017
- Li, Y. G., J. E. Vidale, K. Aki, F. Xu, and T. Burdette (1998b). Evidence of shallow fault zone strengthening after the 1992 M7.5 Landers, California, earthquake, *Science* 279, 217–219.
- Yong-Gang Li, John E. Vidale, Steven M. Day, David D. Oglesby, and Elizabeth Cochran, Postseismic Fault Healing on the Rupture Zone of the 1999 M 7.1 Hector Mine, California, Earthquake, BSSA, 2003.
- Richard J. O'Connell, Bernard Budiansky, Seismic velocities in dry and saturated cracked solids, *JGR*, 1974.
- Qin, L., Ben-Zion, Y., Bonilla, L. F., & Steidl, J. H. (2020). Imaging and monitoring temporal changes of shallow seismic velocities at the Garner Valley near Anza, California, following the M7.2 2010 El Mayor-Cucapah earthquake. *Journal of Geophysical Research: Solid Earth*, 125, e2019JB018070.
- Schaff, D. P., and G. C. Beroza (2004), Coseismic and postseismic velocity changes measured by repeating earthquakes, *J. Geophys. Res.*, 109, B10302
- Stanchits, S., S. Vinciguerra, G. Dresen (2006), Ultrasonic velocities, Acoustic emission characteristics and crack damage of basalt and granite, *Pure Appl. Geophys.*, 163, 1–20, doi:10.1007/s00024-006-0059-5.
- Wang, Q.-Y., and Yao, H. J. (2020). Monitoring of velocity changes based on seismic ambient noise: A brief review and perspective. *Earth Planet. Phys.*, 4(5), 532
- Zhang et al., Spatio-temporal variations of shallow seismic velocity changes in Salton Sea Geothermal Field, California in response to large regional earthquakes and long-term geothermal activities, *Earthquake Research Advances*, 2023
- ZHIGANG PENG, and YEHUDA BEN-ZION, Temporal Changes of Shallow Seismic Velocity Around the Karadere-Düzce Branch of the North Anatolian Fault and Strong Ground Motion, *Pure appl. geophys.* 163 (2006)

I read the author's response and the revised manuscript carefully. However, I still have some concerns on the necessity to average the state evolution for different τ . I believe that the τ_{max} estimated by the interpolated average velocity recovery observed after Tocopilla and Iquique earthquakes (see Figure S4) does not reflect the relaxation timescale of seismic wave velocity after a single earthquake. Because the τ_{max} derived from Figure S4 includes the influence of multiple aftershocks on the recovery process. Aftershocks can cause the seismic wave velocity to take more time to recover to its pre-earthquake level, which means that in reality, the relaxation time of seismic wave velocity after a single earthquake will be much shorter than the fitted value of τ_{max} . However, in Figure 2c of the manuscript, the authors used τ_{max} as the relaxation time after a single earthquake to synthesize the recovery process, which is not very reasonable. I think using a smaller constant relaxation time may also yield a good result. It also should be noted that, as shown in Figure 2c, the seismic wave velocity can be restored to its pre-earthquake level in about 4 years after the Tocopilla and Iquique earthquakes. However, the τ_{max} obtained in the manuscript is much longer than 4 years. From Figure 7 in Illien et al. (2012), it can be easily inferred that the τ_{max} should be smaller than 4 years.

[figure redacted]

[figure redacted]